# Variability and properties of liquid-dominated clouds over the ice-free and sea-ice-covered Arctic Ocean

Marcus Klingebiel[1], André Ehrlich[1], Elena Ruiz-Donoso[1], Nils Risse[2], Imke Schirmacher[2], Evelyn Jäkel[1], Michael Schäfer[1], Kevin Wolf[5], Mario Mech[2], Manuel Moser[3, 4], Christiane Voigt[3, 4], and Manfred Wendisch[1]

[1]Leipziger Institut für Meteorologie (LIM), Universität Leipzig, Leipzig, Germany
[2]Institut für Geophysik und Meteorologie (IGM), Universität zu Köln, Cologne, Germany
[3]Institut für Physik der Atmosphäre, Deutsches Zentrum für Luft- und Raumfahrt, Wessling, Germany
[4]Institut für Physik der Atmosphäre, Johannes Gutenberg-Universität, Mainz, Germany
[5]Institut Pierre-Simon Laplace, Sorbonne Université / CNRS, Paris, France

**Correspondence:** Marcus Klingebiel (marcus.klingebiel@uni-leipzig.de)

**Abstract.** Due to their potential to either warm or cool the surface, liquid-phase clouds and their interaction with the ice-free and sea-ice-covered ocean largely determine the energy budget and surface temperature in the Arctic. Here, we use airborne measurements of solar spectral cloud reflectivity obtained during the ACLOUD campaign in summer 2017 and the AFLUX campaign in spring 2019 in the vicinity of Svalbard to retrieve microphysical properties of liquid-phase clouds. The retrieval was tailored to provide consistent results over sea-ice and open ocean surfaces. Clouds including ice crystals that significantly bias the retrieval results were filtered from the analysis. A comparison with in-situ measurements shows a good agreement with the retrieved effective radii and an overestimation of the liquid water path and a reduced agreement for boundary-layer clouds with varying fractions of ice water content. Considering these limitations, retrieved microphysical properties of clouds observed over ice-free ocean and sea-ice in spring and early summer in the Arctic are compared. In early summer, the liquid-phase clouds have a larger median effective radius (9.5 µm), optical thickness (11.8) and effective liquid water path (72.3 g m$^{-2}$) compared to spring conditions (8.7 µm, 8.3, 51.8 g m$^{-2}$, respectively). The results show larger cloud droplets over the ice-free Arctic Ocean compared to sea-ice in spring and early summer caused mainly by the temperature differences of the surfaces and related convection processes. Due to their larger droplet sizes the liquid clouds over the ice-free ocean have slightly reduced optical thicknesses and lower liquid water contents compared to the sea-ice surface conditions. The comprehensive data set on microphysical properties of Arctic liquid-phase clouds is publicly available and could, e.g., help to constrain models or be used to investigate effects of liquid-phase clouds on the radiation budget.

## 1 Introduction

Over the past three decades, the Arctic region has experienced an enhanced warming, which exceeds the global warming by a factor of 2 to 4 (Serreze and Francis, 2006; Serreze and Barry, 2011; Wendisch et al., 2017; Rantanen et al., 2022). This resulted in a drastic decrease of the Arctic sea-ice extent (e.g., Stroeve et al., 2012), which changes the surface energy budget and surface fluxes of heat and moisture. The intertwined processes and feedback mechanisms behind these and further rapid changes

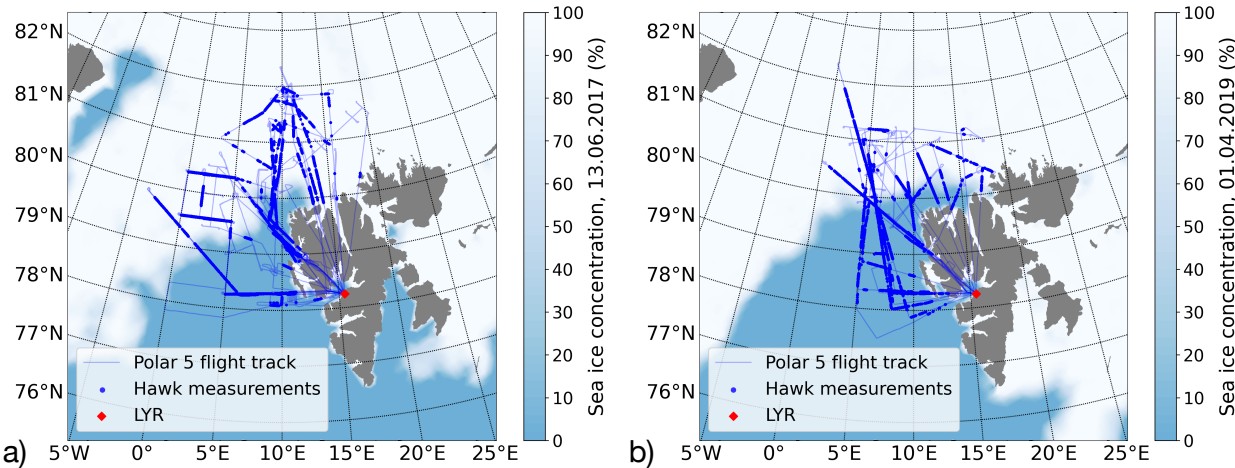

**Figure 1.** Flight tracks of *Polar 5* during the ACLOUD (a) and AFLUX (b) campaign in the vicinity of Svalbard, including Longyearbyen airport (LYR) and the flight sections when the AISA Hawk instrument was measuring. The sea-ice concentration is based on AMSR-E/AMSR2 datasets (Spreen et al., 2008).

of the Arctic climate system are widely referred to as Arctic amplification (Wendisch et al., 2017, 2022a). To investigate and better understand the causes, i.e., the involved key processes and major feedback mechanisms, and effects of Arctic amplification, the Transregional Collaborative Research Center called *Arctic Amplification: Climate Relevant Atmospheric and Surface Processes, and Feedback Mechanisms* ((AC)[3], www.ac3-tr.de) was initiated.

Within the framework of (AC)[3], several field studies were conducted to cover a variety of spatial and temporal scales. In this study, we focus on the airborne campaigns *Arctic CLoud Observations Using airborne measurements during polar Day* (ACLOUD, May/June 2017, Wendisch et al., 2019) and *Arctic Amplification: FLUXes in the Cloudy Atmospheric Boundary Layer* (AFLUX, March/April 2019, Mech et al., 2022), which were performed to study the development of boundary-layer clouds over the sea-ice-covered and ice-free Arctic Ocean. These clouds are often mixed-phase clouds and are suspected to be one of the important factors that contribute to Arctic amplification (Serreze and Barry, 2011), because the partitioning between liquid water droplets and ice crystals within these clouds determines their radiative properties and life cycle (Tan and Storelvmo, 2019).

Commonly, two methods are used to measure the properties of boundary-layer cloud particles from aircraft. First, to sample them directly with in-situ instruments. This has the advantage that the size and shape of individual cloud particles can be measured and the accuracy of the instrument can be estimated by a prior calibration. Second, to use passive or active remote sensing measurements to retrieve the cloud properties. In contrast to in-situ sampling, remote sensing observations cover a larger measurement area. However, the information retrieved from passive remote sensing using reflectances often is dominated by the cloud top properties (Platnick, 2000). Unfortunately, passive remote sensing retrieval from reflectances of Arctic boundary-layer clouds is challenging due to the unknown vertical distribution of ice particles in the typically liquid-dominated

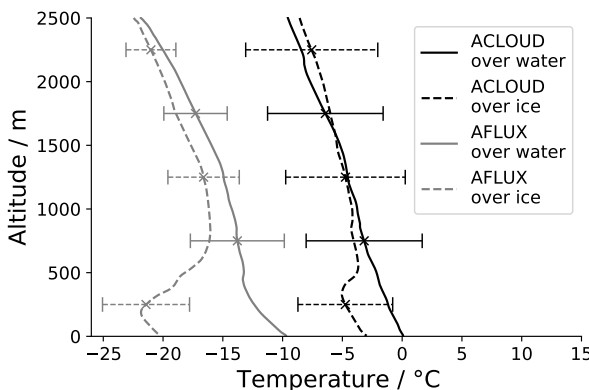

**Figure 2.** Averaged temperature profiles of all launched dropsondes during the ACLOUD (black) and AFLUX (gray) campaign over water (continues lines) and ice surface (dashed). The horizontal bars represent the standard deviation.

clouds (Ruiz-Donoso et al., 2020) and the changing surface albedo differences (i.e., ice-free ocean, sea-ice or snow). Liu et al. (2010) and Ehrlich et al. (2017) showed that the surface albedo and surface emissivity differences between ice-free ocean and sea-ice influence the detection of clouds with passive remote sensing measurements. They compared results from MODIS (passive remote sensing) and CloudSat/CALIPSO (active remote sensing) and found that cloud amount trends in the Arctic show potential differences of up to 3 % per decade. The reason for that is partly the small difference in emissivity between clouds and sea-ice, which often leads to the misclassification of thin clouds as sea-ice.

In this study, we apply a retrieval method of cloud properties which is based on the bi-spectral method proposed by Nakajima and King (1990) and Ruiz-Donoso et al. (2020) to identify the properties of boundary-layer clouds over Arctic sea-ice and ice-free ocean. To minimize the uncertainty of the retrieval method, only liquid-dominated clouds are analysed, which were filtered using the slope-phase index parameter (Ehrlich et al., 2008). The cloud properties derived over sea-ice from the bi-spectral retrieval method suffer from the uncertainties of the assumed sea-ice albedo (e.g., Ehrlich et al., 2017). To reduce this uncertainty, this study makes use of airborne measurements of sea-ice albedo conducted within the campaign, which represent the regional and seasonal sea-ice conditions as close as possible. Comparing clouds over ice-free Arctic Ocean and sea-ice gives an impression of how cloud properties could change in a future ice-free Arctic summer.

The paper is structured as follows: Section 2 introduces the remote-sensing and in-situ instruments. In Sect. 3 we present the retrieval method. Section 4 compares between the retrieved microphysical properties and in-situ measurements. Section 5 shows the differences of the cloud properties over ice-free ocean and sea-ice of the ACLOUD and AFLUX campaign, followed by a discussion of the results.

## 2 Aircraft instrumentation

During the summer campaign, ACLOUD, and the spring campaign, AFLUX, airborne measurements with the *Polar 5* research aircraft (Wesche et al., 2016) were conducted. The *Polar 5* flight tracks of both campaigns are presented in Fig. 1. The aircraft was equipped with a set of passive and active remote sensing instruments as summarized in Ehrlich et al. (2019). Dropsondes were released to deliver vertical profiles of atmospheric parameters. Figure 2 shows the averaged temperature profiles of all launched dropsondes for both campaigns, separated in measurements over sea-ice and open ocean. During AFLUX, cloud probes for in-situ measurements complemented the set-up on the *Polar 5*, while for ACLOUD these measurements were performed with a second aircraft.

### 2.1 SMART Albedometer

The Spectral Modular Airborne Radiation measurement sysTem (SMART) is configured to measure upward and downward spectral solar irradiance from which the albedo in flight altitude is derived. For this purpose, optical inlets are mounted on actively stabilized platforms (Wendisch et al., 2019) and connected via optical fibers to grating spectrometers. The upward and downward irradiance, $F_\lambda^\downarrow$, is measured in a spectral range between 300 nm and 2300 nm with a frequency of 2 Hz and an uncertainty of 8 % (Bierwirth et al., 2013; Wendisch et al., 2019; Ehrlich et al., 2019).

### 2.2 AISA Hawk spectral imager

The Airborne Imaging Spectrometer for Applications (AISA) Hawk instrument (Ruiz-Donoso et al., 2020; Ehrlich et al., 2019) consists of a downward-viewing push broom sensor aligned across the flight track to measure 2D fields of upward radiance. The push broom sensor contains 384 across-track pixels, where each pixel performs spectral measurements between 930 nm and 2550 nm wavelength in 288 channels. With a 36° field of view (FOV) and a sampling frequency of 20 Hz, the instrument has a spatial resolution of 2 m, assuming a distance of 1 km between aircraft and cloud (Ruiz-Donoso et al., 2020). The flight tracks with the location of the measurements from the AISA Hawk spectral imager during the summer (ACLOUD) and spring (AFLUX) campaign are displayed in Fig. 1. Due to storage capacities AISA Hawk data are only recorded when clouds are present below the aircraft. The uncertainty of the measured radiance is approximately 6 % (Schäfer et al., 2013; Ruiz-Donoso et al., 2020).

The spectral reflectivity, $R_\lambda$, is calculated by using the upward radiance measurements, $I_\lambda^\uparrow$, from the AISA Hawk spectral imager combined with the downward spectral irradiance, $F_\lambda^\downarrow$, measurements from the SMART Albedometer:

$$R_\lambda = \pi \cdot \frac{I_\lambda^\uparrow}{F_\lambda^\downarrow}. \tag{1}$$

However, during the AFLUX campaign condensation on the inside of the optics or an improperly working stabilization platform made the downward radiance and irradiance measurements unreliable. For this reason, the SMART measurements during AFLUX are replaced by simulations of $F_\lambda^\downarrow$, which were performed with the Library of Radiative transfer (libRadtran) code

(Mayer and Kylling, 2005; Emde et al., 2016). According to Ehrlich et al. (2023) the accuracy of downward simulations is high as atmospheric conditions measured by radiosondes (Ny-Ålesund) and aerosol optical depth (airborne sun photometer) were implemented in the simulations. Within libRadtran we used the radiative transfer solver DISORT2 (Discrete Ordinate Radiative Transfer, Stamnes et al., 2000) and and performed the simulations of the upward radiance for solar zenith angles between 55° and 69°. Azimuth angles were adjusted depending on measurement time, location and attitude of the research aircraft.

Based on $R_\lambda$, we calculate the slope phase index, $PI$,

$$PI = 100 \cdot \frac{(\lambda_\mathrm{b} - \lambda_\mathrm{a})}{R_{1640}} \left( \frac{\mathrm{d}R_\lambda}{\mathrm{d}\lambda} \right)_{[\lambda_\mathrm{a}, \lambda_\mathrm{b}]} \tag{2}$$

for the spectral reflectivity range between $\lambda_\mathrm{a}$ = 1550 nm and $\lambda_\mathrm{b}$ = 1700 nm. For typical Arctic conditions, a threshold of $PI < 20$ serves as an indicator to identify liquid water clouds and exclude mixed-phase and ice clouds from the analysis when required (Ehrlich et al., 2008). The $PI$ is most sensitive to the amount of ice crystals and liquid water droplets close to the cloud top. Therefore, the clouds identified by the threshold of $PI < 20$ need to be considered as liquid-dominated clouds.

## 2.3 MiRAC cloud radar

The Microwave Radar/radiometer for Acrtic Clouds (MiRAC) combines a frequency-modulated continuous wave (FMCW) radar at 94 GHz including a 89 GHz passive channel (MiRAC-A) and an eight-channel radiometer with frequencies between 175 and 340 GHz (MiRAC-P, Mech et al., 2019). MiRAC is mounted at the bottom of the fuselage of the *Polar 5* with an inclination of ~25° (MiRAC-A) and 0° (MiRAC-P) to study low-level, Arctic mixed-phase clouds. The FMCW radar delivers vertically resolved profiles of equivalent radar reflectivity (Kliesch and Mech, 2019; Mech et al., 2022) for the ACLOUD and AFLUX campaigns.

## 2.4 In-situ cloud particle instruments

During AFLUX, *Polar 5* was equipped with an advanced particle measurement configuration including scattering and optical array probes. The Cloud Aerosol Spectrometer (CAS) uses forward scattered laser light (4-12°, given by the manufacturer) to estimate the cloud droplet size distributions in a diameter size range between 2.8 - 50 µm, based on Mie theory (Wendisch and Brenguier, 2013; Klingebiel et al., 2015; Voigt et al., 2017; Kleine et al., 2018; Voigt et al., 2022). Shadow images of hydrometeors are recorded by two optical array probes, the Cloud Imaging Probe (CIP) and the Precipitation Imaging Probe (PIP, Baumgardner et al., 2001; Klingebiel et al., 2015). Both imaging probes differ in pixel resolution, which results in two different ranges for particle size detection (CIP: Observable size range from 15 - 960 µm; PIP: Observable size range from 100 - 6400 µm). By a combined particle size distribution from all three instruments (CAS, CIP, and PIP), microphysical cloud properties including the effective radius, $r_\mathrm{eff}$, the Liquid Water Content ($LWC$), and the Ice Water Content ($IWC$) is calculated. In this study, the $r_\mathrm{eff}$ calculation is based on all observable cloud particle sizes, the $LWC$ is calculated using particles smaller than 50 µm (CAS data) and $IWC$ using particles larger than 50 µm (CIP and PIP), which is appropriate for Arctic mixed-phase clouds (McFarquhar et al., 2007; Korolev et al., 2017). Uncertainties of in-situ cloud measurements

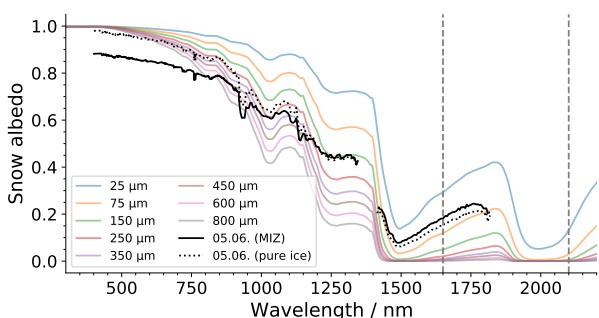

**Figure 3.** Measured (black) and simulated (colored) snow albedo. The measurements were taken during the ACLOUD flight on 5 June 2017 over pure sea-ice (dotted line) and the Marginal sea-Ice Zone (MIZ) (continuous line). The simulations cover grain sizes from 25 µm up to 800 µm and are based on Zege et al. (2011). The vertical dashed lines indicate the wavelengths (1650 nm and 2100 nm) which are used for the retrieval.

strongly depend on the microphysical cloud properties. In liquid clouds, the droplets are sized by the CAS, which has a range of 10-50 % uncertainty (Baumgardner et al., 2017), while in ice and mixed-phase clouds the sizing is dominated by data from the optical array probes which have an uncertainty of 20 % (Baumgardner et al., 2017; Gurganus and Lawson, 2018). In stratiform liquid and mixed phase clouds, the calculation of the LWC is subject to an error of 20 % (Faber et al., 2018) and for the IWC an error of 50 % (Heymsfield et al., 2010; Hogan et al., 2012) is assumed. For the in-situ data used here, a description of the
processing methods and the derivation of microphysical cloud properties are described in detail by Mech et al. (2022) and Moser et al. (2023).

## 3  Retrieval method and design

Cloud microphysical and optical parameters (effective radius, $r_{\text{eff}}$, effective liquid water path, $LWP_{\text{eff}}$, and cloud optical thickness, $\tau$) are retrieved from the AISA Hawk and SMART measurements on the *Polar 5* aircraft, in combination with
130 forward radiative transfer simulations, to generate look-up tables of cloud top reflectivity. The design and limits of the retrieval are demonstrated in the following:

The radiative transfer simulations are performed with the library for Radiative transfer (*libRadtran*) code from Mayer and Kylling (2005); Emde et al. (2016) using the radiative transfer solver DISORT2 (Stamnes et al., 2000). Solar zenith angles (72° to 82° during AFLUX and 55° to 69° during ACLOUD, according to Wendisch et al. (2022b)) and azimuth angles were
135 adjusted for each simulation, depending on the location, altitude, and measurement time of the aircraft.

Due to the low contrast between clouds and bright sea-ice surfaces at visible wavelengths, bi-spectral cloud retrieval typically use measurements at wavelengths larger than 1000 nm (Platnick et al., 2016). For the retrieval method presented here, the reflectivities at 1650 nm and 2100 nm are applied. Following the approach by, e.g., Ehrlich et al. (2017) or Ruiz-Donoso

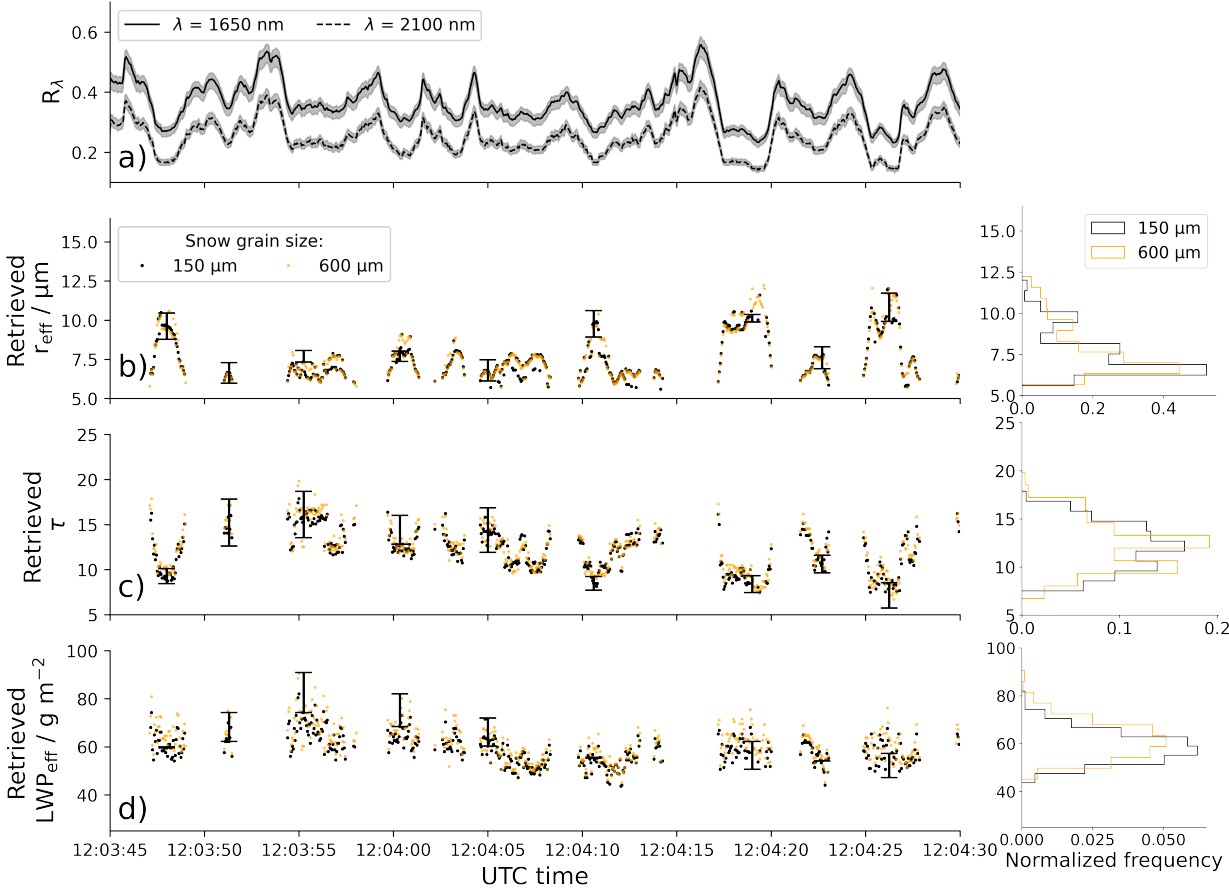

**Figure 4.** (a) Reflectivity, R$_\lambda$, calculated from AISA Hawk radiance along center pixels. The gray shaded area indicates an AISA Hawk uncertainty of $\pm 6\,\%$ (b) Retrieved effective radius, $r_{\text{eff}}$. (c) Retrieved optical thickness, $\tau$. (d) Retrieved effective liquid water path, $LWP_{\text{eff}}$. The error-bars for the retrieved parameters indicate an error propagation for an initial AISA Hawk uncertainty of $\pm 6\,\%$. All observations were taken on 24 March 2019 during the AFLUX campaign.

et al. (2020), the reflectivity at 2100 nm is normalized to a spectral ratio $R_{\text{ratio}}$. To match the observed cloud reflectivities, an appropriate estimate of the surface albedo in the radiative transfer simulations is required.

As shown by Ehrlich et al. (2017), an incorrectly assumed surface albedo can bias the retrieval significantly (Fricke et al., 2014; Platnick, 2001). This holds especially for sea-ice, where the surface albedo may change due to different snow grain sizes, leads, and melt ponds (Jäkel et al., 2019; Jäkel et al., 2021). The albedo measurement was obtained in cloud-free conditions. Because interaction between clouds and surface albedo have to be considered, the measurements cannot be applied directly in the radiative transfer simulations. Instead, a parametrization of the snow albedo for cloudy conditions depending on the snow grain size is used. To identify, which snow grain size represents the ACLOUD conditions, airborne measurements of snow albedo obtained from low-level flights (5 June 2017) are used. The albedo averaged for the marginal sea-ice zone (MIZ, black line) and pure ice (dotted line) are shown in Fig. 3 and compared to theoretical snow albedo derived from the snow albedo model by Zege et al. (2011) for different snow grain sizes, assuming a homogeneous snow profile. It is obvious that the differences between the measurements and the simulations changes spectrally, which might be caused by a non-homogeneous stratification of snow with different grain sizes or a moistening process taken place at the surface, such as melting snow. The latter one seems more likely, because the albedo simulations were only done for dry snow conditions and the measurements are consistent with observations from Light et al. (2022) and Rosenburg et al. (2023).

Figure 4a shows a 45 s sample of the spectral reflectivity from the combined measurements of AISA Hawk and SMART, which is shown for wavelengths 1650 nm and 2100 nm, recorded for 45 s over Arctic sea-ice during the *Polar 5* research flight on 24 March 2019. The spectral radiance along the 10 center across-track-pixels (187 - 197) of the AISA Hawk samples is averaged to obtain upward directed spectral radiance along the time of flight.

Look-up tables of $R_\lambda$ were simulated for a liquid water cloud for different cloud properties, by varying $r_{\text{eff}}$ from 4 to 24 µm and $LWP_{\text{eff}}$ from 3 to 390 g m$^{-2}$. The clouds are assumed to be homogeneous layers and follow the plane-parallel geometry of the radiative transfer model. The retrieval of $r_{\text{eff}}$ from passive remote sensing measurements is most sensitive to the cloud top layer, where absorption by cloud particles lowers the reflected radiance. Contrarily, the scattering information in the measurements, which is linked to the retrieved $LWP_{\text{eff}}$, originates from the entire cloud. Therefore, retrieved cloud properties, especially $LWP_{\text{eff}}$ may depend on the vertical cloud structure. For this reason we use the index eff in $LWP_{\text{eff}}$ to make clear that this is an effective parameter based on passive remote sensing measurements, which might be biased by the vertical cloud structure. The optical thickness, $\tau$, is calculated by using $LWP_{\text{eff}}$ and $r_{\text{eff}}$:

$$\tau = \frac{3}{2} \cdot \frac{LWP_{\text{eff}}}{\rho_w \cdot r_{\text{eff}}} \tag{3}$$

with the density of water, $\rho_w$. The assumption of homogeneous clouds and the neglect of three dimensional (3D) cloud structures follows the approach by Ruiz-Donoso et al. (2020) and is justified, especially when analyzing $LWP_{\text{eff}}$. As shown by Horváth et al. (2014) the 3D radiative effects are less pronounced in the retrieved $LWP_{\text{eff}}$ compared to the optical thickness. Solar zenith angles were adjusted to flight time and location. A simulation was conducted for each AISA Hawk sample. The typical length of a sample was 3.6 minutes for ACLOUD and 2.1 minutes for AFLUX. For the flight section presented in Fig. 4a,

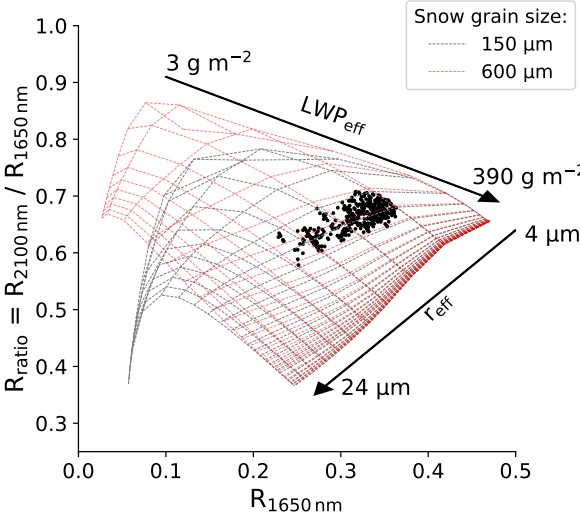

**Figure 5.** Retrieval grids combining the reflectivities at 1650 nm and the ratio of 2100 nm to 1650 nm for the measurement period shown in Fig. 4 from the AFLUX flight on 24 March 2019. The black dots indicate the reflectivities measured by the AISA Hawk (see Eq. 1). The colors show the retrieval grid for a snow grain size of 150 μm (gray) and for 600 μm (red).

the corresponding simulation is displayed as a retrieval grid in Fig. 5, assuming two different grain sizes. The reflectivities used here may still be affected by the variability sea-ice/snow albedo (Ehrlich et al., 2017).

The retrieval uses the reflectivity at 1650 nm wavelength and the reflectivity ratio of 2100 nm to 1650 nm wavelengths. This
wavelength combination was chosen to minimize the dependence of sea-ice and ocean surface albedo, similar to Platnick (2001) and Ehrlich et al. (2017). The use of a ratio instead of a single wavelength reflectivity aims to increase the sensitivity to the cloud particle effective radius and to eliminate potential calibration biases (Werner et al., 2013; Ehrlich et al., 2017). To quantify the effect that different grain sizes of the surface snow layer had on the measurements of AFLUX and ACLOUD, the retrieval was assuming two different snow grain sizes (150 μm and 600 μm). Fig. 4b to Fig. 4d display the retrieved microphysical properties
for a grain size of 150 μm and 600 μm. The difference between both retrieval grids (see Fig. 5) indicates the uncertainties of the cloud retrieval due to the snow grain size. These uncertainties are largest for optically thin clouds with low $LWP$ and increases with droplet size. The set of exemplary measured data is located in a range where both retrieval grids match and the uncertainties due to the assumption of the snow grain size is lower. However, for the observed boundary layer cloud the difference between the retrieved $r_{\mathrm{eff}}$ is 1.2 % on average. It seems that in the selected range of snow grain sizes, the assumption
of snow grain size has only a minor effect on the retrieval. Nevertheless, the closest agreement in Fig. 3 between simulations and measurements occurs for a snow grain size of 150 μm, which was finally selected for the albedo assumed in the retrieval. Since albedo measurements are not available for the AFLUX campaign, we use the same simulated albedo for this campaign as well.

The retrieval is limited by the assumption of pure liquid water clouds. As Arctic boundary-layer clouds are often charac-
terized by a dominant liquid-layer at cloud top, the assumption of liquid clouds might be valid in many cases. Despite this
assumption, the retrieval is applied to all boundary-layer clouds observed during the summer and spring campaign independent
of the cloud phase. Afterwards, to avoid ice dominated cloud sections, we apply different filtering techniques to the retrieved
data, which are described in the following paragraph.

## 3.1 Filtering methods for liquid-dominated clouds

The retrieved microphysical liquid cloud properties over the Arctic sea-ice and the ice-free ocean are often biased by ice
crystals inside the clouds. To identify and neglect the biased measurements from our analysis we apply three different filtering
methods:

1. Equation 2 is applied to all retrieval results. Only for a $PI < 20$, which is an indicator for liquid clouds, the retrieved
   values are accepted for further analysis. This method also removes cloud free sections over sea-ice, because the detection
   of sea-ice results in a high $PI$. Over ocean, cloud free sections are identified and removed when the $LWP_{\text{eff}}$ is lower
   than $3\,\text{g}\,\text{m}^{-2}$.

2. We neglect all reflectivity measurements, which are located outside the retrieval grid. These measurements are clearly
   biased and do not deliver plausible results.

3. We ignore AISA Hawk samples when less than 95 % of the reflectivity measurements are located inside the retrieval
   grid. This method is applied, because ice crystals in clouds do not always move the reflectivity measurements outside
   the retrieval grid but partly shift them inside the retrieval grid. If 95 % of the measurements of an AISA Hawk sample
   are located inside the retrieval grid, it is very likely that the retrieved microphysical properties are not influenced by ice
   crystals. However, the disadvantage is a strong reduction of the data set.

## 4 Retrieval uncertainties

Cloud properties are retrieved for the summer and spring campaign, ACLOUD and AFLUX, respectively. To identify the ac-
curacy of the retrieval, we compare the retrieved cloud properties with in-situ observations using a case study. Furthermore, we
use radar observations to identify for which vertical cloud structure ice crystals potentially contaminate and bias the retrieved
$LWP_{\text{eff}}$.

## 4.1 Comparing retrieved cloud effective radius with in-situ measurements

In-situ measurements obtained during the AFLUX campaign for the flight on 24 March 2019 are used to evaluate the retrieved
$r_{\text{eff}}$ from AISA Hawk measurements. Two flight sections, one over ice-free ocean and one over sea-ice, both obtained shortly
after or before the AISA Hawk measurements are selected. The time series of radar reflectivity and flight altitude are shown in
Fig. 6a together with the sea-ice concentration (dashed line) and the radar reflectivity. AISA Hawk sampled the first cloud top

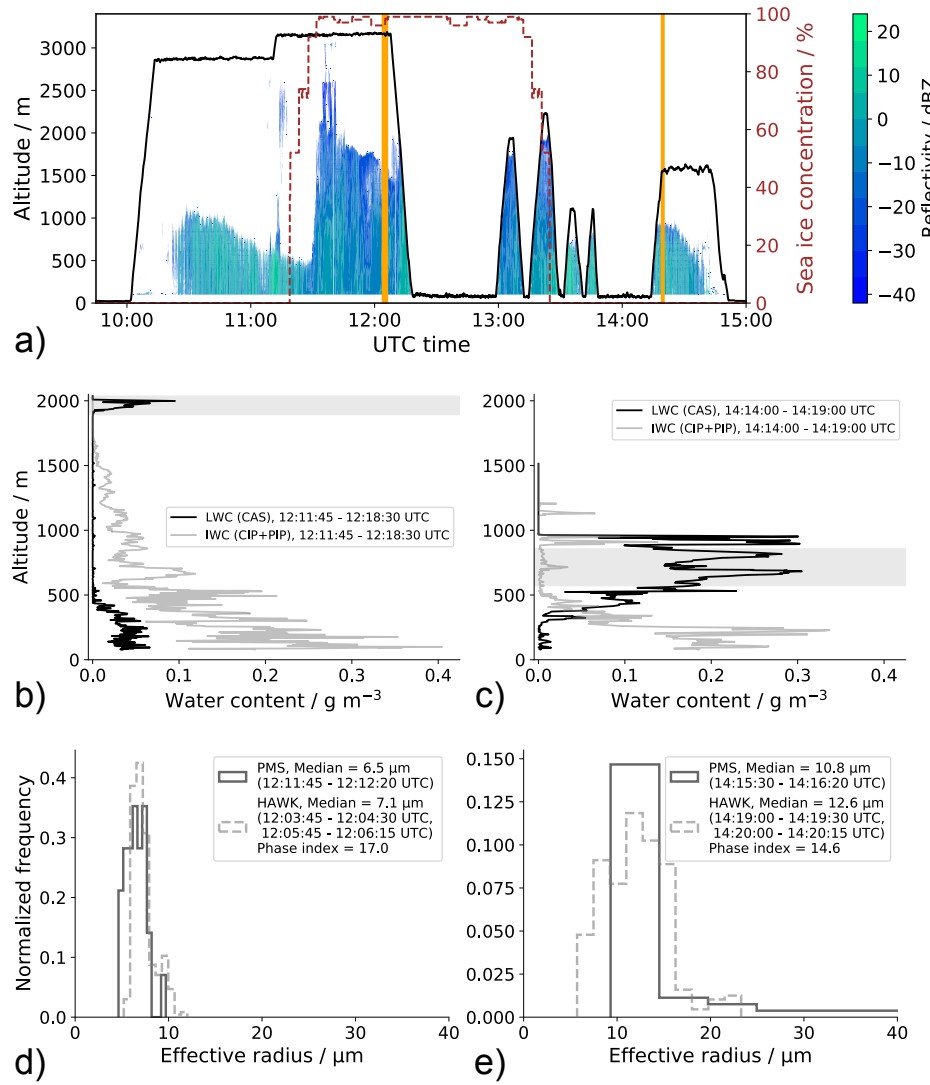

**Figure 6.** a) Sea-ice concentration based on AMSR-E observations (dashed line) and altitude profile (black) of *Polar 5* flight on 24 March 2019 during AFLUX. The vertical lines (orange) indicate the AISA Hawk measurements, which are shown in (d) and (e). The radar reflectivity of the sampled clouds is indicated by the colorbar. b) Measured $LWC$ and $IWC$ profiles from the CAS, CIP and PIP instruments during the decent around 12:10 UTC. c) Same as (b), just for the ascent through the cloud layer around 14:15 UTC. The gray shaded areas mark the sections which were considered for the in-situ measurements of $r_{\mathrm{eff}}$. d) Histogram of the retrieved effective radius for different flight sections, both retrieved over sea-ice, and compared with in-situ measurements. e) Same like in (d), but retrieved and measured over ice-free ocean. The in-situ measurements in (d) and (e) were taken during vertical profiles of the *Polar 5* through the cloud layers.

over an ice surface ($\sim$12:00 UTC, $PI = 17.0$), just before *Polar 5* probed the cloud layer. The $LWC$ profile in Fig. 6b, based on CAS measurements, shows the typical characteristics of clouds over sea-ice, where clouds are driven by cloud top cooling and are located close to the surface, possibly touching the ground. The highest $LWC$ is observed close to the surface.

For the second cloud ($\sim$14:20 UTC, $PI = 14.6$), occurring over an ice-free water surface, *Polar 5* ascended through the cloud layer and conducted in-situ measurements before AISA Hawk sampled the cloud top layer. Over the ice-free ocean (see Fig. 6c), the surface heat fluxes lead to a more convective cloud layer with cloud base at 400 m, and an adiabatic increase of $LWC$ towards cloud top. The in-situ data are filtered for cloud $LWC$ larger than 0.02 g m$^{-3}$ to remove cloud droplets effected by mixing in the entrainment zone (Klingebiel et al., 2015).

The comparison between retrieved $r_{\text{eff}}$ and directly measured $r_{\text{eff}}$ by the in-situ instruments is given for the first and the second cloud case in Fig. 6d and e, respectively. The cloud case over sea-ice, showed a rather complex multi-layer cloud structure. As the retrieval by AISA Hawk is only sensitive to cloud top, only in-situ measurements of $r_{\text{eff}}$ in the upper cloud layer around 2000 m altitude (gray shaded area in Fig. 6b) are used for the comparison. This layer is a pure liquid cloud with small droplet sizes and a median, measured with the in-situ instruments, of 6.5 µm. The lower cloud layers and the sea-ice surface does not significantly affect the AISA Hawk retrieval, which results in only a slightly larger $r_{\text{eff}}$ with a median of 7.1 µm.

While the majority of the effective radii in Fig. 6d occur below 10 µm, the second case is characterized by larger droplets (Fig. 6e). Moreover, the in-situ instruments detect particles in a size bin up to $\sim$60 µm, which indicates the presence of some ice particles. However, also in this case the in-situ and the AISA Hawk measurement show close median values with 10.8 µm and 12.6 µm, respectively. Considering that the AISA Hawk and the in-situ measurements were sampling the same cloud layer at different locations with completely different measurement methods, the comparisons show in both cases (Fig. 6d and Fig. 6e) overlaps with close median values. This indicates, that for typically stratified mixed-phase clouds the assumption of liquid clouds in the retrieval is valid. Filtering with a $PI < 20$ (see section 3a), assures that only such liquid-dominated clouds are analysed. Our presented retrieval method, thus, delivers reasonable results for the estimation of $r_{\text{eff}}$, even though the measurements took place over different surface conditions, namely ice-free ocean and sea-ice, and in the presence of liquid-dominated mixed-phase clouds.

## 4.2 Comparing retrieved optical thickness and liquid water path with in-situ measurements

To evaluate the retrieved $LWP_{\text{eff}}$ based on measurements of AISA Hawk, we integrate the $LWC$ measured by the CAS in-situ instrument between cloud base and cloud top height of the sampled clouds. The vertical $LWC$ profiles of the CAS are presented together with the $IWC$ from the CIP instrument in Fig. 6b and Fig. 6c for the first and second cloud section, respectively. The profiles capture the whole vertical descent and ascent through the clouds, limited only by the minimum flight altitude of 60 m.

Contrarily to the retrieval of $r_{\text{eff}}$, the $LWP_{\text{eff}}$ retrieved from AISA Hawk is an integral value over the entire altitude below the aircraft. The in-situ calculated $LWP$ over sea-ice (Fig. 6b) is 21 g m$^{-2}$. For the cloud over ice-free ocean (Fig. 6c) a value

| | ACLOUD | AFLUX |
|---|---|---|
| **All data** | 1079508 (100 %) | 907091 (100 %) |
| **Data over sea ice surface** | 717857 (66.5 %) | 413982 (45.6 %) |
| **Data over sea ice-free surface** | 361651 (33.5 %) | 493109 (54.4 %) |
| **Data over sea ice surface inside retrieval grid** | 448089 (41.5 %) | 120427 (13.3 %) |
| **Data over sea ice-free surface inside retrieval grid** | 234873 (21.8 %) | 170796 (18.8 %) |

**Table 1.** Number of recorded AISA Hawk data from the ACLOUD and AFLUX campaign. One data point represents one pixel value along the center pixel line of an AISA Hawk sample.

of $101\,\mathrm{g\,m^{-2}}$ is derived. The in-situ $LWP$ for the first section might underestimate the real cloud, because parts of the cloud below the flight altitude were not sampled.

The retrieved $LWP_{\mathrm{eff}}$ averaged over the AISA Hawk samples is $62\,\mathrm{g\,m^{-2}}$ and $119\,\mathrm{g\,m^{-2}}$ ($21\,\mathrm{g\,m^{-2}}$ and $101\,\mathrm{g\,m^{-2}}$ were

measured by the in-situ instruments) for the first and second cloud case, respectively. For the cloud over sea-ice, $LWP_{\mathrm{eff}}$ is overestimated by a factor of three by AISA Hawk. Both cases showed significant $IWP$, which is quantified by the in situ measurements. The retrieval by AISA Hawk assumes the entire cloud to be liquid. Therefore, $IWP$ is to some extend included in the retrieved $LWP_{\mathrm{eff}}$, which will overestimate the in-situ $LWP$. Additionally, a bias in the retrieved $r_{\mathrm{eff}}$ due to the presence of ice particles will lead to an overestimation of $LWP_{\mathrm{eff}}$ by the retrieval. This is a known problem of the retrieval method when

assuming homogeneous liquid vertical profiles and apply it to mixed-phase clouds (Coopman et al., 2019; Ruiz-Donoso et al., 2020). Considering the $IWC$ profiles in Fig. 6b and Fig. 6c, it is obvious that ice particles were present inside the clouds and lead to precipitation (snow) in the lower cloud layers. Only the top layer (around $2000\,\mathrm{m}$ altitude) for the first cloud consisted of pure liquid droplets, which were sampled by the AISA Hawk instrument. All in all, the differences between the measured and retrieved liquid water path seem to be influenced by the distribution of ice particles and liquid water inside the cloud.

To constrain the impact of ice particles on the retrieval biases, the in-situ measurements are converted into extinction profiles of liquid and ice particles following the theory of Eq. 3. The profiles are integrated to the in-situ cloud optical thickness for total, liquid and ice particles. If the extinction by ice particles is low and the extinction by liquid droplets matches the retrieved $\tau$, the observed bias of $LWP_{\mathrm{eff}}$ is mostly due to the assumption of homogeneous clouds. This is the case for the second cloud section, where cloud optical thicknesses of 16.1 (retrieved), 16.35 (in-situ total), 15.97 (in-situ liquid) and 0.37 (in-situ

ice) were derived. For section 1, the comparison does fail (13.2 (retrieved), 2.6 (in-situ total), 1.95 (in-situ liquid, 0.65 (in-situ ice)) due to the mismatch of the cloud location. This is obvious in the radar reflectivity (see Fig.6a), which significantly increases after the remote sensing measurement and while starting the in-situ profile. The high radar reflectivity agrees with the high amount of $IWC$ measured in-situ (Fig.6b). During the AISA Hawk measurements, the radar reflectivity was still lower indicating a more liquid dominated cloud. Unfortunately, this makes a comparison of the $LWP_{\mathrm{eff}}$ and optical thickness

impossible for this section. However, the agreement in retrieved and in-situ $r_{\mathrm{eff}}$, at least indicates, that the liquid cloud top layer did not significantly changed.

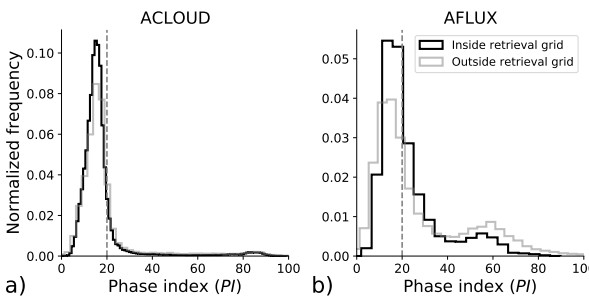

**Figure 7.** Distribution of the $PI$ for the ACLOUD (a) and AFLUX (b) campaign, depending on the location of the AISA Hawk measurements related to the retrieval grid. The vertical dashed lines indicate the threshold, which is be used to discriminate between liquid and mixed-phase clouds.

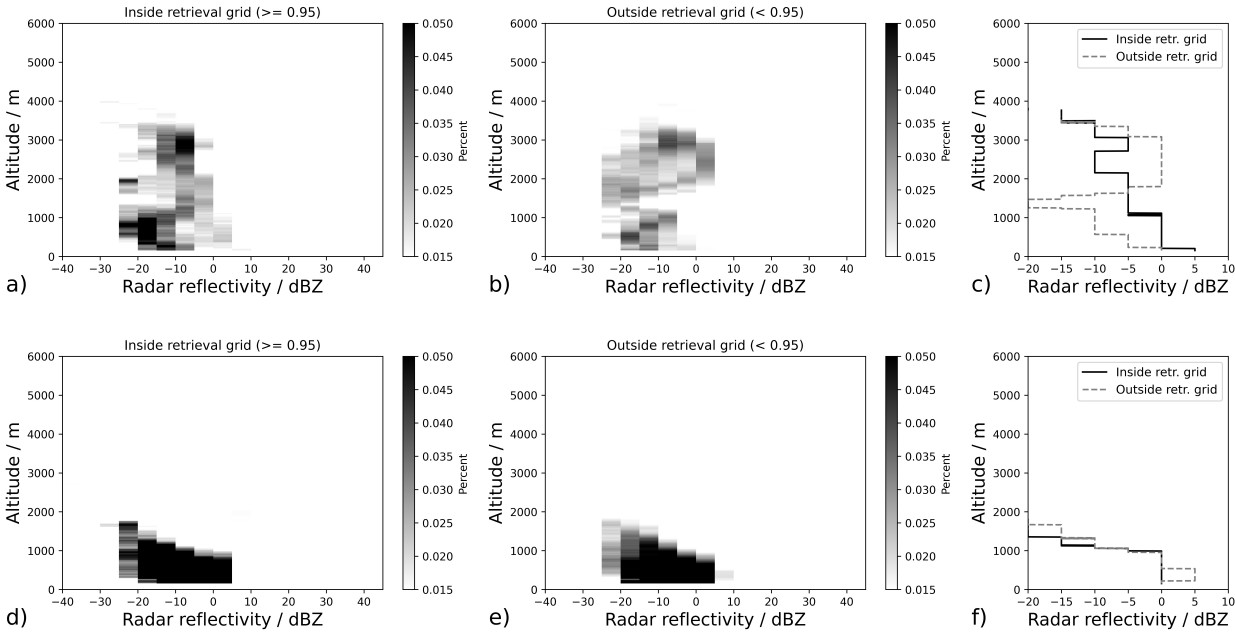

**Figure 8.** a) Contoured frequency by altitude diagram (CFAD) for the ACLOUD campaign, when 95 % of the measurements of each AISA Hawk sample were within the retrieval grid. b) CFAD of the same measurements, but here less than 95 % were within the retrieval grid. c) Profiles of the CFAD plots in (a) and (b), considering the maximum dBZ values with a density > 0.015 %. Plots (d) to (f) show the same, just for the AFLUX campaign.

## 4.3 Statistical evaluation of the retrieval using cloud radar observations

As indicated by the case study, the location of ice particles within the cloud column may bias the retrieved $LWP_{\text{eff}}$. In Arctic mixed-phase clouds, the vertical structure and the amount of ice particles may change on small scales, often linked to updrafts and downdrafts of the cloud (Ruiz-Donoso et al., 2020). A way to quantify how often the retrieval is affected by the presence of ice crystals is to look at the position of the measurements regarding to the retrieval grid. For example, in Fig. 5a all measurements (black dots) are within the retrieval grid which indicates plausible results, even though there are uncertainties for the $LWP_{\text{eff}}$. However, if the observed cloud layer, in particular the cloud top, is dominated by ice particles, then the measurements are affected and might move outside the retrieval grid, like it is shown in Fig. 7b in Ruiz-Donoso et al. (2020). Therefore, the position of the measurements related to the retrieval grid is an indicator of how reliable the retrieved results are.

Nevertheless, unlike the $PI$, this indicator does not provide any information on whether a boundary-layer cloud is dominated by liquid water or ice particles. To make that more clear, Fig. 7 shows for both campaigns the distribution of the $PI$ depending on the position of the measurements inside the retrieval grid. While for the ACLOUD campaign (Fig. 7a) the distributions look similar, they show some differences for the AFLUX campaign (Fig. 7b). As AFLUX was characterized by colder temperatures favoring the presence of mixed-phase clouds, measurements are located more often outside the retrieval grid when the $PI$ is higher. Nevertheless, it becomes clear that the position of the measurements relative to the retrieval grid cannot be used as an indicator of the amount of ice particles inside a boundary-layer cloud, because the $PI$ is not directly linked to the position of the measurements in- and outside the retrieval grid.

To identify how reliable the retrieval results are, we analyse how often the measurements are located inside the retrieval grid for both campaigns. The results are presented in Table 1. For AFLUX, 19 % and 13 % of measurements lie inside the retrieval grid over a sea-ice-free and over a sea-ice surface. During the ACLOUD campaign, 22 % of the measurements were covered by the retrieval grid over a water surface, and up to 42 % over an ice surface. These numbers indicate that less cloud ice particles were present during ACLOUD than during the AFLUX campaign, which is related to the lower atmospheric temperatures (see Fig. 2) because ACLOUD was conducted later in the year.

To estimate the ice concentration that is responsible for a mismatch between the measurements and the retrieval grid, we use the MiRAC cloud radar measurements from both campaigns. The cloud radar is sensitive to ice particles and measures the portion of ice inside the clouds, which is shown by the detected radar reflectivity. Figure 8 shows the distribution of the radar reflectivity for measurements inside and outside the retrieval grid. To avoid retrieval cases with measurements simultaneously inside and outside the retrieval grid we only consider AISA Hawk samples with more than 95 % of the measurements inside the retrieval grid, like it is explained in Sect.3a. Unfortunately, this reduces the amount of data to 7 % for ACLOUD and 2 % for the AFLUX campaign.

Figure 8a shows the highest contribution of ice particles below 1500 m between -20 dBZ and -10 dBZ. These signals are related to snow, which shows values up to 15 dBZ below 1500 m. Between 1500 m and 3500 m the maximum radar reflectivity increases to 0 dBZ. In Fig. 8b, the precipitation below 1500 m is visible as well. Above 1500 m the radar reflectivities show values up to 5 dBZ, which is higher than in Fig. 8a.

| | | All sampled data inside grid | | Samples with 95% of data inside grid | | Difference | |
|---|---|---|---|---|---|---|---|
| | | ACLOUD | AFLUX | ACLOUD | AFLUX | ACLOUD | AFLUX |
| All | PI | 15,1 | 17,9 | 13,4 | 13,5 | 1,7 | 4,4 |
| | $r_{\mathrm{eff}}$ / µm | 9,5 | 8,7 | 9,3 | 8,5 | 0,2 | 0,2 |
| | $\tau$ | 11,8 | 8,3 | 11,5 | 8,4 | 0,3 | -0,1 |
| | $LWP_{\mathrm{eff}}$ / g m$^{-2}$ | 72,3 | 51,8 | 69,2 | 49,1 | 3,1 | 2,7 |
| Over water | PI | 12,8 | 16,5 | 12,4 | 14,4 | 0,4 | 2,1 |
| | $r_{\mathrm{eff}}$ / µm | 9,4 | 9,1 | 9,3 | 11,8 | 0,1 | -2,7 |
| | $\tau$ | 10,4 | 7,1 | 10,1 | 6,1 | 0,3 | 1,0 |
| | $LWP_{\mathrm{eff}}$ / g m$^{-2}$ | 65,8 | 48,0 | 62,6 | 51,8 | 3,2 | -3,8 |
| Over ice | PI | 16,1 | 20,8 | 15,3 | 13,0 | 0,8 | 7,8 |
| | $r_{\mathrm{eff}}$ / µm | 9,5 | 8,1 | 9,5 | 8,0 | 0,0 | 0,1 |
| | $\tau$ | 12,5 | 10,2 | 13,1 | 9,3 | -0,6 | 0,9 |
| | $LWP_{\mathrm{eff}}$ / g m$^{-2}$ | 77,4 | 56,8 | 85,1 | 48,3 | -7,7 | 8,5 |

**Table 2.** Median values of the retrieved cloud properties and the slope phase-index, considering data inside the retrieval grid from all samples (see Fig.9), compared to median values considering only samples with 95 % of the measurements located inside the retrieval grid.

To emphasize the difference between Fig. 8a and Fig. 8b, we show in Fig. 8c the profiles of the CFAD plots from Fig. 8a and Fig. 8b and only consider the maximum dBZ values with a density larger than 0.015 %. Here it is obvious that, when the AISA Hawk measurements end up outside the retrieval grid and the retrieval fails, the radar reflectivities are higher in the cloud top layers (see dashed line between 1500 m and 3500 m). A higher radar reflectivity indicates more ice particles inside the clouds, which affects the retrieval method. It can be concluded that the retrieval fails if the radar reflectivity in the cloud top layer is larger than -5 dBZ.

The same analysis is done for the spring campaign, AFLUX, and the results are presented in Fig. 8d to Fig. 8f. Here, it is noticeable that the cloud top is lower than during the summer campaign, ACLOUD. This agrees with Mioche et al. (2015), who reported lower cloud tops during the winter than during the summer months in the vicinity of Svalbard. For the spring campaign, the retrieval failed more often (see Table 1), which is related to a higher portion of ice particles inside the sampled clouds (see Fig. 7b). Therefore, the results in Fig. 8d to Fig. 8f are not as obvious as for the summer campaign (Fig. 8a to Fig. 8c).

## 5 Comparison of retrieved cloud properties over Arctic sea-ice and ice-free ocean

The retrieval results of both campaigns are compared to each other to identify the differences in the liquid water cloud properties over ice-free ocean and Arctic sea-ice. To make sure that we only consider liquid water clouds, we present for the retrieved parameters clouds with a $PI < 20$ and we neglect AISA Hawk measurements located outside the retrieval grid. These are the filtering methods one and two described in Sect.3a. The results are shown as distribution plots in Fig. 9 and show the $PI$, $r_{\mathrm{eff}}$, $\tau$, and $LWP_{\mathrm{eff}}$.

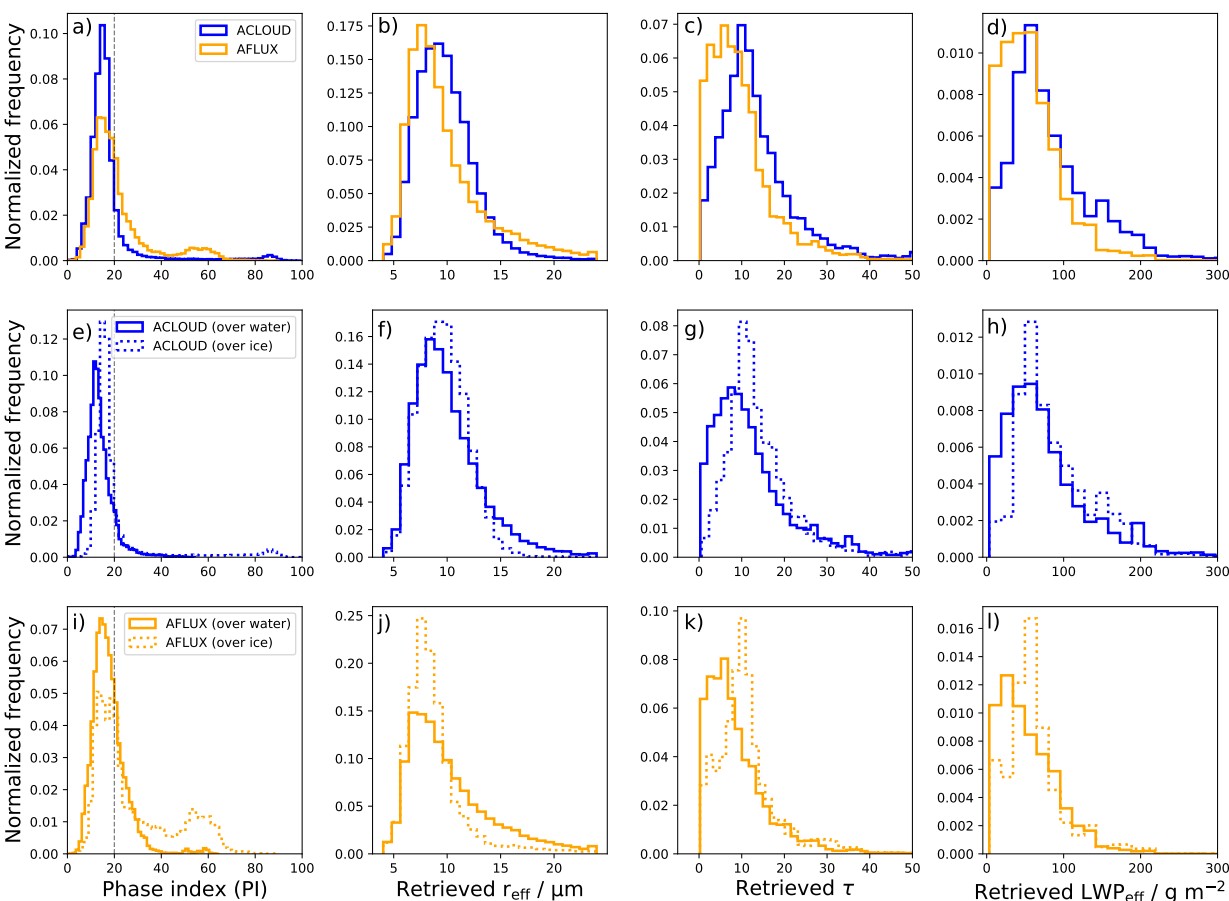

**Figure 9.** Comparison between the early summer, ACLOUD, and the spring, AFLUX, campaign. Panels a - d show the differences of the Phase-Index and the retrieved parameters, $r_{\mathrm{eff}}$, $\tau$ and $LWP_{\mathrm{eff}}$ over ice-free water and sea-ice. Panels e - h and i - l show the same parameters, but exclusively for the ACLOUD and the AFLUX campaign, respectively. Here, the data are separated into measurements over ice-free water (solid lines) and sea-ice (dotted lines).

The third filtering method (samples with less than 95 % of the reflectivity measurements located inside the retrieval grid are ignored) was also applied, but like mentioned before, it reduces the data set to 7 % for ACLOUD and 2 % for AFLUX. Nevertheless, we have a high confidence in the retrieval results shown in Fig. 9 because the differences to the results considering the third filtering method are very minor. This is shown in Table 2 by the median values of the distribution plots for both filtering methods. Moreover, in early summer the liquid-phase clouds have larger median values with $r_{\mathrm{eff}} = 9.5\,\mu\mathrm{m}$, $\tau = 11.8$ and $LWP_{\mathrm{eff}} = 72.3\,\mathrm{g\,m}^{-2}$ in comparison to spring conditions (8.7 μm, 8.3, 51.8 g m$^{-2}$, respectively). That the values of the microphysical cloud properties are larger in summer, is also valid for the observations over water and over sea-ice surface.

The distribution plots of the $PI$ (Fig. 9a) show for the majority of ACLOUD and AFLUX measurements a $PI < 20$ (dashed vertical line), which means that the observed cloud tops were dominated by liquid water droplets. However, observations with $PI > 20$, which represents the occurrence of ice particles or surface ice measured trough optical thin clouds, are common for both campaigns. Two peaks are noticeable around $PI$s of 60 and 85 (see Fig. 9a) and indicate higher in-cloud ice concentrations. Figures 9e and 9i support that these peaks were detected over ice. A closer look at the nadir camera pictures from these flight segments revealed that for the peak detected during AFLUX a haze layer was located over the ice, while for ACLOUD barely any clouds were visible, so that the $PI$ represents in this case the pure ice surface.

Like discussed before, the retrieval method is suited for liquid water clouds and fails when ice particles are present. That is why the clouds are optical thicker in Fig. 9g and k over a sea-ice surface, because optically thin clouds, where the ice on the surface increases the $PI$, are not considered.

The $r_{\mathrm{eff}}$ shows larger values over open ocean than over sea-ice for both campaigns (see Fig. 9f and 9j). This is plausible because the moister and warmer air over open ocean leads to more convection and, therefore, higher cloud tops with larger cloud droplets. Larger cloud droplets over ice-free water consequently lead to smaller optical thickness, $\tau$, which is shown in Fig. 9g and 9k by a shift of the over water modes towards smaller $\tau$.

Focusing on the ice-free ocean, the $LWP_{\mathrm{eff}}$ peaks around 20 g m$^{-2}$ during AFLUX while it peaks around 50 g m$^{-2}$ during ACLOUD. This difference might be related to the seasons. ACLOUD took place roughly two months later in the year with higher temperatures, hence the clouds might have evolved more and produced more liquid water. Despite the different seasons, the $LWP_{\mathrm{eff}}$ over sea-ice is similar for both campaigns, which indicates that over sea-ice the changes might not be that significant than over water.

## 6   Conclusions

Airborne solar spectral radiance measurements from cloud tops are used to retrieve cloud microphysical properties of Arctic liquid-phase clouds during the summer campaign ACLOUD in May/June 2017 and the spring campaign AFLUX in March/April 2019 in the vicinity of Svalbard.

The retrieval method presented in this study is developed for liquid water clouds and, therefore, involves three different filtering techniques to avoid cloud top sections, which are dominated by ice crystals. That the retrieval is applicable for Arctic mixed-phase clouds, which are often covered by a super-cooled liquid cloud top layer, is shown by a comparison of the retrieved

cloud top $r_{\text{eff}}$ with in-situ measurements. In a case study, the retrieved $r_{\text{eff}}$ with median values of 7.1 μm and 12.6 μm showed a good agreement with in-situ measurements with a median value of 6.5 μm and 10.8 μm, respectively, even considering that the measurements were performed successively.

However, the retrieved $LWP_{\text{eff}}$ is overestimated, which is a known issue for this kind of retrieval method and related to ice particles inside the clouds, because the retrieval is suited for pure liquid water clouds.

To identify how much cloud ice the retrieval can tolerate and how the ice needs to be distributed vertically to affect the retrieval results, we compare the retrieval results with radar measurements. The comparison leads to the conclusion that the retrieval method is reliable when the radar reflectivity of the cloud top layer is smaller than -5 dBZ.

Considering these limitations we applied the retrieval method to a data set of airborne measurements of cloud top spectral solar radiances of Arctic boundary-layer clouds to characterize the differences between microphysical properties of clouds observed over ice-free ocean and Arctic sea-ice in spring and early summer in the vicinity of Svalbard. We identified that in early summer the liquid-phase clouds have larger median values with $r_{\text{eff}}$ = 9.5 μm, $\tau$ = 11.8 and $LWP_{\text{eff}}$ = 72.3 g m$^{-2}$ in comparison to spring conditions (8.7 μm, 8.3, 51.8 g m$^{-2}$, respectively). These differences might be related to the temperature

differences between the summer and spring campaign. Independent of the season, the results show larger cloud droplets over the ice-free ocean compared to the Arctic sea-ice. This seems to be caused by the temperature and humidity differences of the surfaces and related convection processes. Because the size of the cloud droplets is larger over the ice-free ocean, $\tau$ and $LWP_{\text{eff}}$ are slightly reduced.

In summary, the presented and comprehensive data set shows the microphysical differences of liquid-phase clouds over

Arctic sea-ice and ice-free ocean for summer and spring conditions. The data are publicly available (Klingebiel et al., 2023a, b) and can be used for studies to constrain models which investigate the effects of Arctic boundary-layer clouds on the radiation budget.

*Data availability.* The here presented comprehensive data sets on microphysical properties of Arctic liquid-phase clouds are published in Klingebiel et al. (2023a) and Klingebiel et al. (2023b). The SMART data used in this study have been published in Jäkel et al. (2019). All

AISA Hawk spectral radiance measurements from the ACLOUD and AFLUX campaigns are published in Ruiz-Donoso et al. (2019) and Schäfer et al. (2021). The measurements from the in-situ instruments were published by Moser and Voigt (2022). The flight track data are from Ehrlich et al. (2018) and Lüpkes et al. (2019).

*Author contributions.* MK is the primary author of the paper. AE, ERD, EJ, MS, KW, MM, CV and MW carried out the airborne experimental work. Simulations with *libRadtran* were performed by ERD, KW and MK. MK and MM compared the retrieval results with in-situ

measurements. NR, IS and MK evaluated the accuracy of the retrieval results with radar observations. All the authors contributed to the interpretation of the results and wrote the paper.

*Competing interests.* The authors declare that they have no conflict of interest.

*Acknowledgements.* Scientifc support was given by Anna E. Luebke. Special thanks to the whole research team, including the engineers and pilots from the ACLOUD and AFLUX campaign. We gratefully acknowledge the funding by the Deutsche Forschungsgemeinschaft (DFG, German Research Foundation) – Projektnummer 268020496 – TRR 172, within the Transregional Collaborative Research Center "ArctiC Amplification: Climate Relevant Atmospheric and SurfaCe Processes, and Feed- back Mechanisms (AC)[3]".

*Financial support.* This research has been supported by the Deutsche Forschungsgemeinschaft (grant no. 268020496 – TRR 172). Funded by the Open Access Publishing Fund of Leipzig University supported by the German Research Foundation within the program Open Access Publication Funding.

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
