# Peer review of "Variability and properties of liquid-dominated clouds over the ice-free and sea-ice-covered Arctic Ocean"

_Atmospheric Chemistry and Physics, 2022_

## Author Comment (AC1)

**Response to the reviewer comments on the manuscript:**
**"Variability and properties of liquid-dominated clouds over the ice-free and sea-ice-covered Arctic Ocean"**
**[acp-2022-848]**

We thank the two anonymous reviewers for diligently reading and carefully reviewing our manuscript and providing us with useful comments and suggestions to improve the quality of the manuscript. A list of all reviewer comments and questions (written in *italics*) as well as our response (written in regular) is given below. Whenever we provide information in which line changes were made we refer to the line numbering of the revised manuscript.

**Comments Reviewer 1:**

*This paper presents a fairly straightforward application of a bispectral retrieval for liquid cloud optical properties from airborne data in the arctic. There is a limited validation of the retrieval against in-situ probe data. It is found that the effective radius agrees well with the in-situ data whereas the liquid water path can be overestimated, presumably due ot the presence of ice. The retrieval is applied to two deployments in different seasons and the optical properties from two seasons are contrasted. I have a few minor comments listed below and two more significant comments.*

*Significant comments:*

*First, I would like to see the optical depth retreival validated to explain the liquid water path biases.*

We appreciate this suggestion to explore the bias caused by ice crystals on the retrieved liquid water path (LWP) in a more quantitative manner. However, we would like to clarify that our study focuses specifically on liquid-phase clouds, and our retrieval method is designed to avoid cloud top sections dominated by ice crystals.

Anyway, we investigated the possibility to include the vertical distribution of ice particles into our libRatran simulations to reduce the bias in the retrieved LWP. The challenge we encountered was that, apart from comparing the results with in-situ measurements, we could not identify the quantity and location of ice particles within the clouds using our passive remote sensing technique for all the measurements conducted during the AFLUX and ACLOUD campaigns. Nevertheless, to ensure transparency in our manuscript and emphasize that the retrieved LWP is solely based on radiative properties, we made the decision to rename it as the retrieved effective liquid water path, $LWP_{eff}$, throughout the entire document. To convey this change, we modified the following description:

Line 163: "For this reason we use the index eff in $LWP_{eff}$ to make clear that this is an effective parameter based on passive remote sensing measurements, which might be biased by the vertical cloud structure."

*Second, I believe there is lidar data for these flights and I would like to see the lidar data compared to bispectral retreivals in addition to the radar since the two instruments together provide a more complete picture of hydrometeor phase throughout the vertical profile.*

Thanks for this comment. Regarding your comment we spend some time to look into these data and implemented the data in the analysis. First, we identified the cloud top heights from the AMALi Lidar and the MiRAC Radar measurement and then calculated their relative distance, which can be used as an indicator of the presence or absence of ice crystals in the cloud top layer. To estimate the difference in cloud top height we interpolated the datasets to a time resolution of 1 Hz. The resulting difference in cloud top heights represents the liquid layer thickness at cloud top. Figure 1 (see below) shows two histograms showing the difference of cloud top height (Lidar minus Radar) for the ACLOUD and AFLUX campaign. It is obvious that more ice particles were present close to cloud top during AFLUX (Fig. 1b) than during ACLOUD. This agrees with our previous results written in the manuscript.

[Figure]

*Figure 1: Difference in cloud top height, based on AMALi Lidar and MiRAC Radar measurements for the ACLOUD (a) and AFLUX (b) campaign.*

Anyway, to see what difference a consideration of the liquid layer depth at cloud top makes for the retrieval results, we applied it to Figure 9 from in the manuscript and the result is shown here in Figure 2 (see below). Here, we applied a further filter in comparison to the manuscript, considering only retrieval results where the liquid layer at cloud top is larger than 10 m. As comparison, the retrieved parameters from the manuscript are plotted in grey (Fig. 2f-h and j-l). As you can see, the differences are hardly noticeable.

Unfortunately, the lidar and radar are not available for all flights and flight sections covered by the AISA Hawk data set. The combined data set is reduced tremendously. For this reason and in light of the good agreement between the filtering method applied in the manuscript and the Radar/Lidar filter, we decided to use the full data set and not implement the additional filter.

[Figure]

*Figure 2: Same as Figure 9 in the manuscript, but with an additional filter. Here, we plot only retrieval results where the liquid layer at cloud top is larger than 10 m. For the retrieved values over ice and water surface (f – h and j – l), we plotted in grey the values from Figure 9 in the manuscript.*

*Minor comments:*

*Line 86: please describe the simulations of the spectral flux.*
We added following information:
Line 91: "Within libRadtran we used the radiative transfer solver DISORT2 (Discrete Ordinate Radiative Transfer, Stamnes et al., 2000) and performed the simulations of the upward radiance for solar zenith angles between 55° and 69°. Azimuth angles were adjusted depending on measurement time, location and attitude of the research aircraft."

*Line 90: Can you provide some estimate of uncertainty in the phase identification either from the references or collocated measurements (e.g. lidar? or in-situ probes). What is the False Alarm Rate, Probability of Detection etc.?*
Thanks for this comment. The phase index calculated from spectral reflectivity is not always unambiguous and therefore there is uncertainty in the phase detection. It can become ambiguous when only small ice crystals are present as shown by Ehrlich et al. (2008). However, the observed clouds were clearly liquid dominated, which was indicated by the presence of glories during the flights. Our filtering aims at removing clouds where the ice crystals can significantly affect the retrieval, e.g., the liquid cloud top layer is not present. Such cases can well be identified by the phase index. As demonstrated above for another question, we checked the performance of the phase index filter by radar/lidar observations of

the cloud top liquid layer. This showed that the current filtering method is sophisticated enough and an additional filtering based on lidar/radar measurements is not necessary.

*Figure 3: you should spell out MIZ in the figure caption.*
We changed it.

*Line 150: neglection -> neglect*
Done.

*Line 188: does -> do*
Changed.

*Line 215: obtains -> results*
Thanks, we changed it.

*Equation 4: how are zbase and ztop chosen? Is this the entire profile or only the liquid cloud layer at the top of the profile?*
You actually can see the chosen part in Fig. 6b and 6c. The top and base of the profiles of the in-situ measured LWC were used for the $z_{top}$ and $z_{base}$. The gray shaded areas mark only the sections which we considered for the in-situ estimation of $r_{eff}$. To make that more clear we wrote:
Line 249: "The profiles capture the whole vertical descent and ascent through the clouds, limited only by the minimum flight altitude of 60 m. The top and base altitude of these *LWC* profiles were used for the estimation of $z_{top}$ and $z_{base}$".

*Line 233- 236: This paragraph is not written clearly. Please rewrite it for clarity. What do you mean by both cloud layers? Do you mean that the integral is over the entire liquid water content profile? In lIne 235, what does the 'first section' refer to?*
Thanks for pointing that out. You are right that does not make sense. It seems like the sentence is a leftover from an earlier iteration. We removed this sentence.

*Section 4.2: You have the measurements to be more quantitative with regard to the bias caused by ice crystals on you retrieved LWP. You should convert the liquid and ice drop size distributions to optical extinctions. Then you can integrate the liquid + ice extinction and compare that with your retrieved optical depths. This will allow you to demonstrate to what extent the scattering by ice crystals is biasing your LWP. You should include profile plots of the calculated liquid and ice extinction.*
Because this comment is about the LWP bias as well, we repeat our response regarding your first comment:
We appreciate this suggestion to explore the bias caused by ice crystals on the retrieved liquid water path (LWP) in a more quantitative manner. However, we would like to clarify that our study focuses specifically on liquid-phase clouds, and our retrieval method is designed to avoid cloud top sections dominated by ice crystals.
Anyway, we investigated the possibility to include the vertical distribution of ice particles into our libRatran simulations to reduce the bias in the retrieved LWP. The challenge we encountered was that, apart from comparing the results with in-situ measurements, we could not identify the quantity and location of ice particles within the clouds using our passive remote sensing technique for all the measurements conducted during the AFLUX and

ACLOUD campaigns. Nevertheless, to ensure transparency in our manuscript and emphasize that the retrieved LWP is solely based on radiative properties, we made the decision to rename it as the retrieved effective liquid water path, $LWP_{eff}$, throughout the entire document. To convey this change, we modified the following description:

Line 163: "For this reason we use the index eff in $LWP_{eff}$ to make clear that this is an effective parameter based on passive remote sensing measurements, which might be biased by the vertical cloud structure."

*Section 4.3 From what I can tell from Wendisch, 2019 it seems like there was an airborne lidar flying as well. You should be able to get a much more precise idea of cloud top phase using the lidar LDR and backscatter.*
We answered that comment already above (second major comment, page 2).

*Figure 9: change a-e -> a-d.*
Thanks, done.

**References:**

Baumgardner, D., Abel, S. J., Axisa, D., Cotton, R., Crosier, J., Field, P., Gurganus, C., Heymsfield, A., Korolev, A., Krämer, M., Lawson, P., McFarquhar, G., Ulanowski, Z., and Um, J.: Cloud Ice Properties: In Situ Measurement Challenges, Meteorological Monographs, 58, 9.1–9.23, https://doi.org/10.1175/amsmonographs-d-16-0011.1, 2017

Donth, T., Jäkel, E., Ehrlich, A., Heinold, B., Schacht, J., Herber, A., Zanatta, M., and Wendisch, M.: Combining atmospheric and snow radiative transfer models to assess the solar radiative effects of black carbon in the Arctic, Atmos. Chem. Phys., 20, 8139–8156, https://doi.org/10.5194/acp-20-8139-2020, 2020.

Ehrlich, A., Bierwirth, E., Wendisch, M., Gayet, J.-F., Mioche, G., Lampert, A., and Heintzenberg, J.: Cloud phase identification of Arctic boundary-layer clouds from airborne spectral reflection measurements: test of three approaches, Atmospheric Chemistry and Physics, 8, 7493–7505, https://doi.org/10.5194/acp-8-7493-2008, 2008.

Ehrlich, A., Zöger, M., Giez, A., Nenakhov, V., Mallaun, C., Maser, R., Röschenthaler, T., Luebke, A. E., Wolf, K., Stevens, B., and Wendisch, M.: A new airborne broadband radiometer system and an efficient method to correct dynamic thermal offsets, Atmospheric Measurement Techniques, 16, 1563–1581, https://doi.org/10.5194/amt-16-1563-2023, 2023.

Faber, S., French, J. R., and Jackson, R.: Laboratory and in-flight evaluation of measurement uncertainties from a commercial Cloud Droplet Probe (CDP), Atmospheric Measurement Techniques, 11, 3645–3659, https://doi.org/10.5194/amt-11-3645-2018, 2018.

Gurganus, C. and Lawson, P.: Laboratory and Flight Tests of 2D Imaging Probes: Toward a Better Understanding of Instrument Performance and the Impact on Archived Data, Journal of Atmospheric and Oceanic Technology, 35, 1533–1553, https://doi.org/10.1175/jtech-d-17-0202.1, 2018.

Heymsfield, A. J., Schmitt, C., Bansemer, A., and Twohy, C. H.: Improved Representation of Ice Particle Masses Based on Observations in Natural Clouds, Journal of the Atmospheric Sciences, 67, 3303–3318, https://doi.org/10.1175/2010jas3507.1, 2010.

Hogan, R. J., Tian, L., Brown, P. R. A., Westbrook, C. D., Heymsfield, A. J., and Eastment, J. D.: Radar Scattering from Ice Aggregates Using the Horizontally Aligned Oblate Spheroid Approximation, Journal of Applied Meteorology and Climatology, 51, 655–671, https://doi.org/10.1175/jamc-d-11-074.1, 2012.

Horváth, , Seethala, C., and Deneke, H.: View angle dependence of MODIS liquid water path retrievals in warm oceanic clouds, Journal of Geophysical Research: Atmospheres, 119, 8304–8328, https://doi.org/https://doi.org/10.1002/2013JD021355, 2014.

Korolev, A., McFarquhar, G., Field, P. R., Franklin, C., Lawson, P., Wang, Z., Williams, E., Abel, S. J., Axisa, D., Borrmann, S., Crosier, J., Fugal, J., Krämer, M., Lohmann, U., Schlenczek, O., Schnaiter, M., and Wendisch, M.: Mixed-Phase Clouds: Progress and Challenges, Meteorological Monographs, 58, 5.1–5.50, https://doi.org/10.1175/amsmonographs-d-17-0001.1, 2017

LeBlanc, S. E., Pilewskie, P., Schmidt, K. S. and Coddington, O.: A spectral method for discriminating thermodynamic phase and retrieving cloud optical thickness and effective radius using transmitted solar radiance spectra, Atmos. Meas. Tech., 8(3), 1361–1383, doi:10.5194/amt-8-1361-2015, 2015.

Light, B., Smith, M. M., Perovich, D. K., Webster, M. A., Holland, M. M., Linhardt, F., Raphael, I. A., Clemens-Sewall, D., Macfarlane, A. R., Anhaus, P., and Bailey, D. A.: Arctic sea ice albedo: Spectral composition, spatial heterogeneity, and temporal evolution observed during the MOSAiC drift, Elementa: Science of the Anthropocene, 10, https://doi.org/10.1525/elementa.2021.000103, 000103, 2022.

McBride, P. and Schmidt, K.: A spectral method for retrieving cloud optical thickness and effective radius from surface-based transmittance measurements, Atmos Chem …, 11(14), 7235–7252, doi:10.5194/acp-11-7235-2011, 2011.

McFarquhar, G. M., Zhang, G., Poellot, M. R., Kok, G. L., McCoy, R., Tooman, T., Fridlind, A., and Heymsfield, A. J.: Ice properties of single-layer stratocumulus during the Mixed-Phase Arctic Cloud Experiment: 1. Observations, Journal of Geophysical Research, 112, https://doi.org/10.1029/2007jd008633, 2007.

Mech, M., Ehrlich, A., Herber, A., Lüpkes, C., Wendisch, M., Becker, S., Boose, Y., Chechin, D., Crewell, S., Dupuy, R., Gourbeyre, C., Hartmann, J., Jäkel, E., Jourdan, O., Kliesch, L.-L., Klingebiel, M., Kulla, B. S., Mioche, G., Moser, M., Risse, N., Donoso, E. R., Schäfer, M., Stapf, J., and Voigt, C.: MOSAiC-ACA and AFLUX - Arctic airborne campaigns characterizing the exit area of MOSAiC, in preparation, 2022

Moser, M., Voigt, C., Jurkat-Witschas, T., Hahn, V., Mioche, G., Jourdan, O., Dupuy, R., Gourbeyre, C., Schwarzenboeck, A., Lucke, J., Boose, Y., Mech, M., Borrmann, S., Ehrlich, A., Herber, A., Lüpkes, C., and Wendisch, M.: Microphysical and thermodynamic phase analyses of Arctic low-level clouds measured above the sea ice and the open ocean in spring and summer, Atmospheric Chemistry and Physics, 23, 7257–7280, https://doi.org/10.5194/acp-23-7257-2023, 2023.

Platnick, S.: Vertical photon transport in cloud remote sensing problems, J. Geophys. Res., 105(D18), 22919–22935, 2000.

Rosenburg, S., Lange, C., Jäkel, E., Schäfer, M., Ehrlich, A., and Wendisch, M.: Retrieval of snow layer and melt pond properties on Arctic sea ice from airborne imaging spectrometer observations, Atmos. Meas. Tech. Discuss. [preprint], https://doi.org/10.5194/amt-2023-64, in review, 2023.

Ruiz-Donoso, E., Ehrlich, A., Schäfer, M., Jäkel, E., Schemann, V., Crewell, S., Mech, M., Kulla, B. S., Kliesch, L.-L., Neuber, R., and Wendisch, M.: Small-scale structure of thermodynamic phase in Arctic mixed-phase clouds observed by airborne remote sensing during a cold air outbreak and a warm air advection event, Atmospheric Chemistry and Physics, 20, 5487–5511, https://doi.org/10.5194/acp-20-5487-2020, 2020.

Schäfer, M., Bierwirth, E., Ehrlich, A., Jäkel, E. and Wendisch, M.: Airborne observations and simulations of three-dimensional radiative interactions between Arctic boundary layer clouds and ice floes, Atmos. Chem. Phys., 15(14), 8147–8163, doi:10.5194/acp-15-8147-2015, 2015.

Smith, W. L., Hansen, C., Bucholtz, A., Anderson, B. E., Beckley, M., Corbett, J. G., Cullather, R. I., Hines, K. M., Hofton, M., Kato, S., Lubin, D., Moore, R. H., Rosenhaimer, M. S., Redemann, J., Schmidt, S., Scott, R., Song, S., Barrick, J. D., Blair, J. B., Bromwich, D. H., Brooks, C., Chen, G., Cornejo, H., Corr, C. A., Ham, S. H., Kittelman, A. S., Knappmiller, S., LeBlanc, S., Loeb, N. G., Miller, C., Nguyen, L., Palikonda, R., Rabine, D., Reid, E. A., Richter-Menge, J. A., Pilewswskie, P., Shinozuka, Y., Spangenberg, D., Stackhouse, P., Taylor, P., Thornhill, K. L., Van Gilst, D. and Winstead, E.: Arctic radiation-icebridge sea and ice experiment: The Arctic radiant energy system during the critical seasonal ice transition, Bull. Am. Meteorol. Soc., 98(7), 1399–1426, doi:10.1175/BAMS-D-14-00277.1, 2017.

Stamnes, K., Tsay, S.-C., Wiscombe, W., and Laszlo, I.: DISORT, a General-Purpose Fortran Program for Discrete-Ordinate-Method Radiative Transfer in Scattering and Emitting Layered Media: Documentation of Methodology, Tech. rep., Dept. of Physics and Engineering Physics, Stevens Institute of Technology, Hoboken, NJ 07030, 2000

Warren, S. G.: Can black carbon in snow be detected by remote sensing?, J. Geophys. Res., 118, 779–786, https://doi.org/10.1029/2012JD018476, 2013. a, b

Wendisch, M., Stapf, J., Becker, S., Ehrlich, A., Jäkel, E., Klingebiel, M., Lüpkes, C., Schäfer, M., and Shupe, M. D.: Effects of variable, ice-ocean surface properties and air mass transformation on the Arctic radiative energy budget, Atmospheric Chemistry and Physics Discussions, 2022, 1–31, https://doi.org/10.5194/acp-2022-614, 2022b.

---

## Author Comment (AC2)

**Response to the reviewer comments on the manuscript:**
**"Variability and properties of liquid-dominated clouds over the ice-free and sea-ice-covered Arctic Ocean"**
**[acp-2022-848]**

We thank the two anonymous reviewers for diligently reading and carefully reviewing our manuscript and providing us with useful comments and suggestions to improve the quality of the manuscript. A list of all reviewer comments and questions (written in *italics*) as well as our response (written in regular) is given below. Whenever we provide information in which line changes were made we refer to the line numbering of the revised manuscript.

**Comments Reviewer 2:**

*Summary:*

*This manuscript presents results from arctic airborne campaigns (ACLOUD and AFLUX), where they measured low arctic clouds over sea ice and open sea. The manuscript is very well written and is quite pertinent to ACP, particularly with respect to advances in the Arctic low cloud, which remains highly difficult to measure.*

*This is great manuscript to read, however there are a few minor comments to address, mostly on some clarification of some points (see list below). After these minor comments are addressed, it is recommended for publication in ACP.*

*General Comments:*

1. *There is combined measurements of in situ cloud drop/ice crystal sizes and remote sensing measurements. While this may be outside the scope of the paper, at least a mention on the actual shape of the size distribution should be included. It would be interesting to see how that matches the commonly expected gamma distribution, with alpha =7 that are typically used in Nakajima & King bi-spectral retrievals for quantifying the effective radius.*
   The shape of particle size distributions are shown and discussed in Moser et al. (2023), which we refer to in the manuscript. We think that a detailed examination of size distributions is not necessary for this manuscript. The size distribution shape has only a minor impact on the cloud radiative properties compared to the cloud phase and LWC.

2. *The date format for ACP is dd month yyyy, e.g., 25 July 2007. There are a few instances of a varying date format.*
   We went trough the manuscript and changed all the dates to the right format.

3. *The spectral slope in the measured snow albedo leaves to believe that there may be other factors, like haze or aerosol layer, present. While that may be not so important given much of the remote sensing is focused on the near infrared regions, at least a mention of the haze/aerosol conditions should be made, and if available more details*

*on how that would impact these retrievals. A note that haze was present is found later in manuscript (line 313). Potential impact of this layer should be explored*

When performing the Albedo measurements, the research aircraft was flown at an altitude of 150 meters above ground and below clouds, which makes the presence of a haze layer very unlikely. It is more probable that there was a moistening process taking place at the surface, such as melting snow, as the Albedo simulations were only conducted for dry snow conditions. This observation is consistent with the melting snow conditions shown in Figure 2 of Light et al. (2022).

We now mention that in the manuscript:

Line 149: "It is obvious that the differences between the measurements and the simulations change spectrally, which might be caused by a non-homogeneous stratification of snow with different grain sizes or a moistening process taken place at the surface, such as melting snow. The latter one seems more likely, because the albedo simulations were only done for dry snow conditions and the measurements are consistent with observations from Light et al. (2022) and Rosenburg et al. (2023)."

*Specific Comments:*

1. *Line 24: What is (TR 172)? If it is a reference, then it is not in the reference list.*
   It describes the project number. To keep it simple, we removed it here and just mention it in the acknowledgments.

2. *Line 38: Please add the caveat that cloud top properties is from passive remote sensing from reflectances, not all passive remote sensing techniques, see Platnick 2000. Additionally, some active (lidar) techniques are also limited to the topmost portion of the cloud. There are transmitted-light based passive remote sensing that have a more even distribution of sampling through the cloud. e.g., McBride et al., 2011, LeBlanc et al., 2015, and Smith et al., 2017*
   Thanks for pointing that out. We changed this sentence and the following sentence to:
   Line 38: "However, the information retrieved from passive remote sensing using reflectances often is dominated by the cloud top properties (Platnick, 2000). Unfortunately, passive remote sensing retrieval from reflectances of Arctic boundary-layer clouds is challenging due to the unknown vertical distribution of ice particles in the typically…"

3. *Figure 2: For that many drop sondes, one wonders how representative are these averages? What is the deviation to the median, and the standard deviation?*
   We added horizontal bars of the standard deviation to the plot, which represents the variability of the dropsonde measurements (see Figure 3, below).

[Figure]

*Figure 1: Averaged temperature profiles of all launched dropsondes during the ACLOUD (black) and AFLUX (gray) campaign over water (continues lines) and ice surface (dashed). The horizontal bars represent the standard deviation.*

4. *Line 73: I'm not certain that the reference to the AISA Hawk instrument requires the book Pu 2017.*
   Agreed. We replaced it with Ruiz-Donoso et al., 2020.

5. *Line 77-79: Why is there missing measurements? Instrument issues, lack of cloud, or measurement quality is not sufficient?*
   The files, which the AISA Hawk instrument produces are very large and need a lot of storage capacity. Therefore, we only start recording when Polar 5 is flying above clouds. For this reason, we don't record data e.g., inside clouds or above land surface. To make that more clear we added:
   Line 80: "Due to storage capacities AISA Hawk data are only recorded when clouds are present below the aircraft."

6. *Line 84: How accurate are the simulated downwelling irradiance? Did you remove the conditions with high clouds? What were the sun angles modeled?*
   Yes, the downward simulations can and were only used when no clouds were present above the aircraft. In these cases, the accuracy of simulations is high for the downward irradiance as atmospheric conditions measured by radiosondes (Ny Alesund) and aerosol optical depth (airborne sun photometer) were implemented in the simulations. An accuracy analysis of airborne measured downward irradiances is discussed by Ehrlich et al. (2023).
   We added following sentences:
   Line 89: "According to Ehrlich et al. (2023) the accuracy of downward simulations is high as atmospheric conditions measured by radiosondes (Ny-Alesund) and aerosol optical depth (airborne sun photometer) were implemented in the simulations. Within libRadtran we used the radiative transfer solver DISORT2 (Discrete Ordinate Radiative Transfer, Stamnes et al. 2000) and performed the simulations of the upward radiance for solar zenith angles between 55° and 69°. Azimuth angles were adjusted depending on measurement time, location and attitude of the research aircraft."

7. *Section 2.4: What is the expected uncertainty in the combined in situ cloud probes for effective radius, LWC and IWC?*
To describe the uncertainty of the in-situ instruments we adapted the paragraph to:
Line 117: "In this study, the reff calculation is based on all observable cloud particle sizes, the LW C is calculated using particles smaller than 50 μm (CAS data) and IWC using particles larger than 50 μm (CIP and PIP), which is appropriate for Arctic mixed-phase clouds (McFarquhar et al., 2007; Korolev et al., 2017). Uncertainties of in-situ cloud measurements strongly depend on the microphysical cloud properties. In liquid clouds, the droplets are sized by the CAS, which has a range of 10-50 % uncertainty (Baumgardner et al., 2017), while in ice and mixed-phase clouds the sizing is dominated by data from the optical array probes which have an uncertainty of 20 % (Baumgardner et al., 2017; Gurganus and Lawson, 2018). In stratiform liquid and mixed phase clouds, the calculation of the LWC is subject to an error of 20 % (Faber et al., 2018) and for the IWC an error of 50 % (Heymsfield et al., 2010; Hogan et al., 2012) is assumed. For the in-situ data used here, a description of the processing methods and the derivation of microphysical cloud properties are described in detail by Mech et al. (2022) and Moser et al. (2023)."

8. *Line 121: How low were the sun angles? Arctic often suffers from sun being near the horizon which are hard to model and measure.*
We changed this sentence to the following and added some information:
Line 133: "Solar zenith angles (72° to 82° during AFLUX and 55° to 69° during ACLOUD, according to Wendisch et al. (2022b)) and azimuth angles were adjusted for each simulation, depending on the location, altitude, and measurement time of the airborne measurements."

9. *Line 137: The spectral shape of the measurements vs the modeled snow albedo, particularly in the visible, (shorter wavelength range), seems to indicate that there is something else ins the measurement scene that is not accounted for by the model. Is there any indication of aerosol near surface? Additionally, there may be issues with the Langley scattering in the modeled radiances. At the very least, please explain why you have solely attributed the differences to snow grain size and the stratification.*
The aerosol conditions during the flights were rather clean as indicated by Lidar and sun photometer measurements. This makes it unlikely, that aerosol particles will have impacted the surface albedo measurements. Aerosol particles sedimented into the snow are known to have significantly smaller effect (Donth et al., 2020, Warren 2013). When performing the Albedo measurements, the research aircraft was flown at an altitude of 150 meters above ground and below clouds, which makes the presence of a haze layer very unlikely. It is more probable that there was a moistening process taking place at the surface, such as melting snow, as the Albedo simulations were only conducted for dry snow conditions. This observation is consistent with the melting snow conditions shown in Figure 2 of Light et al. (2022). We now mention that in the manuscript:
Line 149 :"It is obvious that the differences between the measurements and the simulations change spectrally, which might be caused by a non-homogeneous stratification of snow with different grain sizes or a moistening process taken place at the surface, such as melting snow. The latter one seems more likely, because the

albedo simulations were only done for dry snow conditions and the measurements are consistent with observations from Light et al. (2022) and Rosenburg et al. (2023)."

10. *Line 150-152: This is good to identify potential 3D radiative transfer issues, however the abstract and other sections of the text do not make such a distinction, and presents the effective radius and LWP as equally valid. Maybe some bounding of the expected error for Ref, tau, and LWP should be mentioned. A citation might be all that is needed, like Schäfer et al., 2015.*
Good point, we added following reference:
Line 168: "As shown by Horvath et al. (2014) the 3D radiative effects are less pronounced in the retrieved $LWP_{reff}$ compared to the optical thickness."

11. *Line 215: grammar error: "obtaines in"*
We changed it to "results in".

12. *Figure 8 gives a great statement to how well the filtering process is successful.*
Thanks, we like it too.

13. *Line 277: How many days/cases does the 2% of the data represent?*
The two percent represent different sections over several days, from 23 March 2019 to 11 April 2019.

14. *Line 355: many question marks: bad format or is the author unsured that the document is in preparation?*
Thanks for seeing this. The question marks were a reminder that we need to put the right reference here, what we missed to do for the initial submission. We wanted to wait to publish the dataset until we got the reviews for this manuscript, in case we needed to change anything in the processing method.

15. *Data availability: There is no link to the access of the data, but rather a list of papers that describe it.*
The references Klingebiel et al. (2023a) and Klingebiel et al. (2023a) will be published at PANGAEA and link to the dataset.

**References:**

Baumgardner, D., Abel, S. J., Axisa, D., Cotton, R., Crosier, J., Field, P., Gurganus, C., Heymsfield, A., Korolev, A., Krämer, M., Lawson, P., McFarquhar, G., Ulanowski, Z., and Um, J.: Cloud Ice Properties: In Situ Measurement Challenges, Meteorological Monographs, 58, 9.1–9.23, https://doi.org/10.1175/amsmonographs-d-16-0011.1, 2017

Donth, T., Jäkel, E., Ehrlich, A., Heinold, B., Schacht, J., Herber, A., Zanatta, M., and Wendisch, M.: Combining atmospheric and snow radiative transfer models to assess the solar radiative effects of black carbon in the Arctic, Atmos. Chem. Phys., 20, 8139–8156, https://doi.org/10.5194/acp-20-8139-2020, 2020.

Ehrlich, A., Bierwirth, E., Wendisch, M., Gayet, J.-F., Mioche, G., Lampert, A., and Heintzenberg, J.: Cloud phase identification of Arctic boundary-layer clouds from airborne spectral reflection measurements: test of three approaches, Atmospheric Chemistry and Physics, 8, 7493–7505, https://doi.org/10.5194/acp-8-7493-2008, 2008.

Ehrlich, A., Zöger, M., Giez, A., Nenakhov, V., Mallaun, C., Maser, R., Röschenthaler, T., Luebke, A. E., Wolf, K., Stevens, B., and Wendisch, M.: A new airborne broadband radiometer system and an efficient method to correct dynamic thermal offsets, Atmospheric Measurement Techniques, 16, 1563–1581, https://doi.org/10.5194/amt-16-1563-2023, 2023.

Faber, S., French, J. R., and Jackson, R.: Laboratory and in-flight evaluation of measurement uncertainties from a commercial Cloud Droplet Probe (CDP), Atmospheric Measurement Techniques, 11, 3645–3659, https://doi.org/10.5194/amt-11-3645-2018, 2018.

Gurganus, C. and Lawson, P.: Laboratory and Flight Tests of 2D Imaging Probes: Toward a Better Understanding of Instrument Performance and the Impact on Archived Data, Journal of Atmospheric and Oceanic Technology, 35, 1533–1553, https://doi.org/10.1175/jtech-d-17-0202.1, 2018.

Heymsfield, A. J., Schmitt, C., Bansemer, A., and Twohy, C. H.: Improved Representation of Ice Particle Masses Based on Observations in Natural Clouds, Journal of the Atmospheric Sciences, 67, 3303–3318, https://doi.org/10.1175/2010jas3507.1, 2010.

Hogan, R. J., Tian, L., Brown, P. R. A., Westbrook, C. D., Heymsfield, A. J., and Eastment, J. D.: Radar Scattering from Ice Aggregates Using the Horizontally Aligned Oblate Spheroid Approximation, Journal of Applied Meteorology and Climatology, 51, 655–671, https://doi.org/10.1175/jamc-d-11-074.1, 2012.

Horváth, , Seethala, C., and Deneke, H.: View angle dependence of MODIS liquid water path retrievals in warm oceanic clouds, Journal of Geophysical Research: Atmospheres, 119, 8304–8328, https://doi.org/https://doi.org/10.1002/2013JD021355, 2014.

Korolev, A., McFarquhar, G., Field, P. R., Franklin, C., Lawson, P., Wang, Z., Williams, E., Abel, S. J., Axisa, D., Borrmann, S., Crosier, J., Fugal, J., Krämer, M., Lohmann, U., Schlenczek, O., Schnaiter, M., and Wendisch, M.: Mixed-Phase Clouds: Progress and Challenges, Meteorological Monographs, 58, 5.1–5.50, https://doi.org/10.1175/amsmonographs-d-17-0001.1, 2017

LeBlanc, S. E., Pilewskie, P., Schmidt, K. S. and Coddington, O.: A spectral method for discriminating thermodynamic phase and retrieving cloud optical thickness and effective radius using transmitted solar radiance spectra, Atmos. Meas. Tech., 8(3), 1361–1383, doi:10.5194/amt-8-1361-2015, 2015.

Light, B., Smith, M. M., Perovich, D. K., Webster, M. A., Holland, M. M., Linhardt, F., Raphael, I. A., Clemens-Sewall, D., Macfarlane, A. R., Anhaus, P., and Bailey, D. A.: Arctic sea ice albedo: Spectral composition, spatial heterogeneity, and temporal evolution observed during the MOSAiC drift, Elementa: Science of the Anthropocene, 10, https://doi.org/10.1525/elementa.2021.000103, 000103, 2022.

McBride, P. and Schmidt, K.: A spectral method for retrieving cloud optical thickness and effective radius from surface-based transmittance measurements, Atmos Chem …, 11(14), 7235–7252, doi:10.5194/acp-11-7235-2011, 2011.

McFarquhar, G. M., Zhang, G., Poellot, M. R., Kok, G. L., McCoy, R., Tooman, T., Fridlind, A., and Heymsfield, A. J.: Ice properties of single-layer stratocumulus during the Mixed-Phase Arctic Cloud Experiment: 1. Observations, Journal of Geophysical Research, 112, https://doi.org/10.1029/2007jd008633, 2007.

Mech, M., Ehrlich, A., Herber, A., Lüpkes, C., Wendisch, M., Becker, S., Boose, Y., Chechin, D., Crewell, S., Dupuy, R., Gourbeyre, C., Hartmann, J., Jäkel, E., Jourdan, O., Kliesch, L.-L., Klingebiel, M., Kulla, B. S., Mioche, G., Moser, M., Risse, N., Donoso, E. R., Schäfer, M., Stapf, J., and Voigt, C.: MOSAiC-ACA and AFLUX - Arctic airborne campaigns characterizing the exit area of MOSAiC, in preparation, 2022

Moser, M., Voigt, C., Jurkat-Witschas, T., Hahn, V., Mioche, G., Jourdan, O., Dupuy, R., Gourbeyre, C., Schwarzenboeck, A., Lucke, J., Boose, Y., Mech, M., Borrmann, S., Ehrlich, A., Herber, A., Lüpkes, C., and Wendisch, M.: Microphysical and thermodynamic phase analyses of Arctic low-level clouds measured above the sea ice and the open ocean in spring and summer, Atmospheric Chemistry and Physics, 23, 7257–7280, https://doi.org/10.5194/acp-23-7257-2023, 2023.

Platnick, S.: Vertical photon transport in cloud remote sensing problems, J. Geophys. Res., 105(D18), 22919–22935, 2000.

Rosenburg, S., Lange, C., Jäkel, E., Schäfer, M., Ehrlich, A., and Wendisch, M.: Retrieval of snow layer and melt pond properties on Arctic sea ice from airborne imaging spectrometer observations, Atmos. Meas. Tech. Discuss. [preprint], https://doi.org/10.5194/amt-2023-64, in review, 2023.

Ruiz-Donoso, E., Ehrlich, A., Schäfer, M., Jäkel, E., Schemann, V., Crewell, S., Mech, M., Kulla, B. S., Kliesch, L.-L., Neuber, R., and Wendisch, M.: Small-scale structure of thermodynamic phase in Arctic mixed-phase clouds observed by airborne remote sensing during a cold air outbreak and a warm air advection event, Atmospheric Chemistry and Physics, 20, 5487–5511, https://doi.org/10.5194/acp-20-5487-2020, 2020.

Schäfer, M., Bierwirth, E., Ehrlich, A., Jäkel, E. and Wendisch, M.: Airborne observations and simulations of three-dimensional radiative interactions between Arctic boundary layer clouds and ice floes, Atmos. Chem. Phys., 15(14), 8147–8163, doi:10.5194/acp-15-8147-2015, 2015.

Smith, W. L., Hansen, C., Bucholtz, A., Anderson, B. E., Beckley, M., Corbett, J. G., Cullather, R. I., Hines, K. M., Hofton, M., Kato, S., Lubin, D., Moore, R. H., Rosenhaimer, M. S., Redemann, J., Schmidt, S., Scott, R., Song, S., Barrick, J. D., Blair, J. B., Bromwich, D. H., Brooks, C., Chen, G., Cornejo, H., Corr, C. A., Ham, S. H., Kittelman, A. S., Knappmiller, S., LeBlanc, S., Loeb, N. G., Miller, C., Nguyen, L., Palikonda, R., Rabine, D., Reid, E. A., Richter-Menge, J. A., Pilewswskie, P., Shinozuka, Y., Spangenberg, D., Stackhouse, P., Taylor, P., Thornhill, K. L., Van Gilst, D. and Winstead, E.: Arctic radiation-icebridge sea and ice experiment: The Arctic radiant energy system during the critical seasonal ice transition, Bull. Am. Meteorol. Soc., 98(7), 1399–1426, doi:10.1175/BAMS-D-14-00277.1, 2017.

Stamnes, K., Tsay, S.-C., Wiscombe, W., and Laszlo, I.: DISORT, a General-Purpose Fortran Program for Discrete-Ordinate-Method Radiative Transfer in Scattering and Emitting Layered Media: Documentation of Methodology, Tech. rep., Dept. of Physics and Engineering Physics, Stevens Institute of Technology, Hoboken, NJ 07030, 2000

Warren, S. G.: Can black carbon in snow be detected by remote sensing?, J. Geophys. Res., 118, 779–786, https://doi.org/10.1029/2012JD018476, 2013.  a, b

Wendisch, M., Stapf, J., Becker, S., Ehrlich, A., Jäkel, E., Klingebiel, M., Lüpkes, C., Schäfer, M., and Shupe, M. D.: Effects of variable, ice-ocean surface properties and air mass transformation on the Arctic radiative energy budget, Atmospheric Chemistry and Physics Discussions, 2022, 1–31, https://doi.org/10.5194/acp-2022-614, 2022b.

---

## Author Response (AR2)

**Response to the reviewer comment on the manuscript:**
**"Variability and properties of liquid-dominated clouds over the ice-free and sea-ice-covered Arctic Ocean"**
**[acp-2022-848]**

**Comment Reviewer 1:**

*I do not feel the authors adequately addressed my comment regarding the optical depth. Perhaps I did not explain my thoughts thoroughly enough.*
*I suspect that while the retrieved liquid water path is biased high relative to the in-situ data, the optical depth retrieval probably agrees relatively well with the combined (liquid+ice) optical depth.*
*You should be able to calculate the in-situ extinction coefficient for both ice and liquid using the measured drop size distributions and then integrate vertically to estimate the combined optical depth.*
*This is a simple calculation - I don't think there is any need to add ice particles to your look up tables and include them in the bi-spectral retrieval.*
*If you perform this comparison of measured optical depth and in-situ optical depth and they agree, then this provides some physical justification for the assertion that the presence of ice is the primary cause for the biases in the retrieved liquid water path. This would greatly strengthen the paper.*

Thanks for the detailed question. We have put some effort in your suggested calculation and present the results in the following.

[Figure]

*Figure 1: Profiles of the extinction coefficient over sea-ice (a) and ice-free ocean (b). Both plots show the extinction coefficients for in-situ measurements from CAS and CIP/PIP. The total optical depth is provided in the legend.*

Your suggestion to compare the optical thickness, $\tau$, of the retrieved and in-situ measurements is great. To do so, we first looked up the retrieved optical thickness for the two cases presented in Figure 6 in the manuscript, which are:

Section 1 (over sea-ice):  $\tau_{retr.(sea-ice)} = 13.2$

Section 2 (over ice-free ocean):  $\tau_{retr.,(ocean)} = 16.1$

The goal is to show if the retrieved optical thickness matches the in-situ measurements. If they agree, then it would provide a physical justification that the presence of ice is the primary

cause for the biases in the liquid water path, like you suggested.

To calculate $\tau_{in-situ}$ we use

$$\tau_{in-situ} \ = \ \int_{z=0}^{h} b_{ext}(z) \, dz \tag{1}$$

$$\text{with} \quad b_{ext}(z) \ = \ \frac{3}{2} \frac{CWC(z)}{\rho \; r_{eff}(z)} \,. \tag{2}$$

The parameter ρ describes the density of water (for CAS, ice for CIP/PIP). The *CWC* is either the *LWC* or the *IWC*, depending on the in-situ instrument. *CWC* and $r_{eff}$ result from the vertical profiles of the in-situ measurements.

For the single layer cloud of section 2, over the ice-free ocean, the optical thickness between retrieved and in-situ data almost perfectly match with $\tau_{retr.,(ocean)}$ = 16.1 and $\tau_{in-situ}$ = 16.35 $(CAS + CIP/PIP)$. This result shows clearly that in this case the ice particles do not significantly contribute to the total optical thickness. A bias in in-situ observed and retrieved *LWP*$_{eff}$ results mainly from the retrieval assumption of a homogenous cloud. This confirms your assumption. Thanks again!

For section 1, the comparison does fail due to the mismatch of the cloud location. This is obvious in the radar reflectivity which significantly increases after the remote sensing measurements and while starting the in-situ profile (see Figure 6 in the manuscript).

Of course, we want to mention this result in the manuscript and therefore we changed section 4.2 in the manuscript (see track-changes file) and added the following paragraph (line 265 to 276):

"To constrain the impact of ice particles on the retrieval biases, the in-situ measurements are converted into extinction profiles of liquid and ice particles following the theory of Eq. 3. The profiles are integrated to the in-situ cloud optical thickness for total, liquid and ice particles. If the extinction by ice particles is low and the extinction by liquid droplets matches the retrieved τ, the observed bias of *LWP*$_{eff}$ is mostly due to the assumption of homogeneous clouds. This is the case for the second cloud section, where cloud optical thicknesses of 16.1 (retrieved), 16.35 (in-situ total), 15.97 (in-situ liquid) and 0.37 (in-situ ice) were derived. For section 1, the comparison does fail (13.2 (retrieved), 2.6 (in-situ total), 1.95 (in-situ liquid, 0.65 (in-situ ice)) due to the mismatch of the cloud location. This is obvious in the radar reflectivity, which significantly increases after the remote sensing measurement and while starting the in-situ profile. The high radar reflectivity agrees with the high amount of *IWC* measured in-situ. During the AISA Hawk measurements, the radar reflectivity was still lower indicating a more liquid dominated cloud. Unfortunately, this makes a comparison of the *LWP*$_{eff}$ and optical thickness impossible for this section. However, the agreement in retrieved and in-situ r$_{eff}$, at least indicates, that the liquid cloud top layer did not significantly changed."